# CoRAL: Contact-Rich Adaptive LLM-based Control for Robotic Manipulation

## Abstract

Vision-Language-Action (VLA) systems often lack adaptability and explainability due to their black-box structure and dependency on fixed action sets from extensive tele-operated datasets, limiting their effectiveness in complex, dynamic manipulation scenarios. To address this issue, we propose Contact-Rich Adaptive LLM-based Control (CoRAL), a novel modular framework capable of effectively managing complex, dynamic, and contact-rich manipulation tasks. By integrating foundational vision and language models with motion planning and reactive controllers, our system achieves zero-shot planning and adaptive manipulation without relying on extensive tele-operated action datasets. Unlike conventional VLAs, we explicitly separate the roles of vision models and Large Language Models (LLM): the vision module handles environmental parameter initialization and object pose tracking, while the LLM generates initial contact strategies and cost function estimations. This collaboration establishes a physical understanding of the scene, instantiated as a dynamic planning world model for our planner. Additionally, this modular approach significantly enhances both the explainability and performance of the overall framework, as demonstrated by ablation studies. Furthermore, we introduce a memory unit to leverage past manipulation experiences, enabling the generalization and efficient reuse of learned contact strategies and parameter adjustments across diverse manipulation scenarios. Experiments conducted on challenging contact-rich tasks validate our framework's robustness and highlight the critical design elements that contribute to its effectiveness. Website: *https://sites.google.com/view/CoRAL*

## 1 Introduction

Foundational models have demonstrated significant success in various fields, leading to increased efforts to apply these models within robotics (Firoozi et al., 2025; Tayyab Khan & Waheed, 2025). Particularly, Vision-Language-Action (VLA) systems have garnered considerable attention for their potential in robotic manipulation tasks (Ma et al., 2024; Zhong et al., 2025; Sapkota et al., 2025). However, existing VLA frameworks struggle to effectively handle contact-rich manipulation tasks, which constitute a substantial portion of daily interactions (Hao et al., 2025; Yu et al., 2025; Xue et al., 2025). These tasks pose significant challenges, as they require not only precise trajectory planning but also sophisticated interaction force management and adaptive control strategies. Achieving success in such complex scenarios typically necessitates extensive training through teleoperation or detailed dynamic modeling, methods that are labor-intensive and reduce generalizability.

Humans, by contrast, rely on initial estimations, subsequently refine their strategies based on sensory feedback, and adjust interactions accordingly (Flanagan et al., 2006; Johansson & Flanagan, 2009; Kim et al., 2015). Similar to this cognitive framework, we propose a novel modular system, **Contact-Rich Adaptive LLM-based Control (CoRAL)**, that integrates reasoning, planning, and control modules into a cohesive architecture. Our model begins by estimating 6-DoF object poses from RGB-D data using FoundationPoseWen et al. (2024), and then a Vision-Language Model (VLM) infers physical parameters such as mass and friction from the estimated object poses, the environment image, and the textual task description (Fig. 1). The planning stage generates initial contact strategies and actions, which are executed in the evaluation environment through reactive control modules. The outcomes from these actions are continuously monitored, with the tracked poses from FoundationPose being used for iterative refinement of plans.

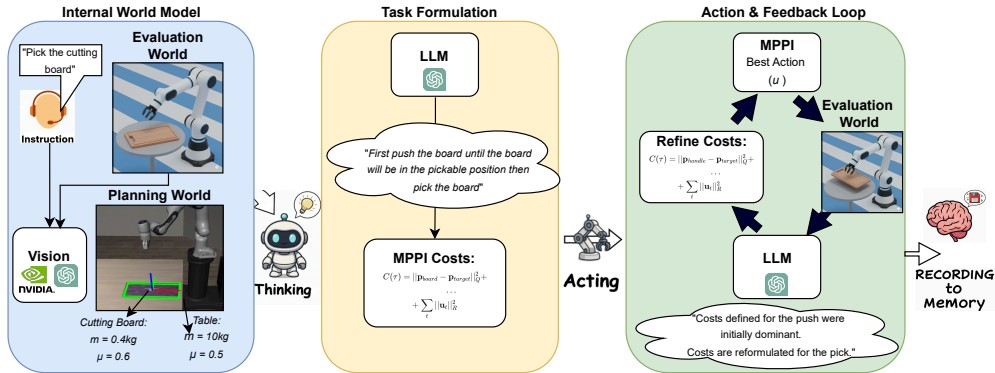

Figure 1: The conceptual workflow of CoRAL, illustrated with the "pick the cutting board" task. Photorealistic images were synthetically generated.

A key innovation of our approach is the strategic integration of vision and Large Language Models (LLMs) with motion planners and controllers, substantially enhancing action explainability and enabling more effective reasoning. Our modular structure clearly delineates roles: vision module manages parameter estimation for environment modeling, while LLM provides symbolic reasoning, initial contact strategies, and cost estimations. The reactive controller then applies these symbolic outputs in the evaluation world, establishing a tight feedback loop between high-level strategy and low-level sensory information. Our main contributions are:

- We propose a novel framework that integrates a Large-Language-Model (LLM) model with a reactive motion controller, enabling zero-shot planning and robust execution for dynamic, contact-rich manipulation.

- Unlike monolithic VLA architectures, we explicitly separate the roles of the vision models for perception and the LLM for reasoning, a design choice that demonstrably enhances both manipulation performance and system explainability.

- We introduce an LLM-driven, closed-loop feedback mechanism that enables the system to adapt its plan mid-execution, successfully completing complex, multi-step manipulation sequences.

- Our framework's performance and adaptability are further enhanced by a memory unit that stores and retrieves past experiences to bootstrap effective solutions for novel tasks.

We evaluate CoRAL on a challenging suite of manipulation tasks, including novel contact-rich problems, such as picking up a thin object from a table and standardized benchmarks from the LIBERO suite Liu et al. (2023). Our extensive experiments and detailed ablation studies confirm that this modular structure significantly improves system performance and explainability, effectively addressing the limitations of conventional VLA frameworks.

## 2 RELATED WORK

**From End-to-End Policies to Decoupled Reasoning** The advent of foundation models has shifted robotic manipulation towards Vision-Language-Action (VLA) models, which learn general-purpose policies from large-scale datasets Firoozi et al. (2025); Ma et al. (2024). Leading examples like OpenVLA Kim et al. (2025b), $\pi_0$ Black et al. (2024) and RT-X O'Neill et al. (2024) utilize an end-to-end approach, directly mapping multimodal inputs to low-level actions. While powerful, this paradigm's reliance on imitation learning makes it data-dependent and often brittle in novel physical scenarios, particularly those involving complex contact dynamics not well-represented in the training data. To overcome these limitations, an emerging trend decouples high-level reasoning from low-level control. Frameworks like ThinkAct Huang et al. (2025), Inner Monologue Huang et al. (2022) and those using Embodied Chain-of-Thought (ECoT) Zawalski et al. (2024) leverage LLMs to generate explicit reasoning steps that guide a separate, learned action policy. Similarly,

MolmoAct Lee et al. (2025) produces mid-level spatial plans as "trajectory traces" before predicting actions, enhancing explainability and steerability. The OneTwoVLA architecture Lin et al. (2025) formalizes this by explicitly modeling a System 1 (fast, reactive acting) and System 2 (slow, deliberate reasoning). Our work, CoRAL, aligns with this decoupling philosophy but takes a distinct neuro-symbolic path by grounding the LLM's reasoning directly in a controller, rather than another learned model.

**Integrating Foundation Models with Motion Planners and Controllers** A promising alternative to end-to-end learning is the integration of foundation models with traditional motion planners, leveraging semantic understanding to guide physically-grounded trajectory optimization. For instance, IMPACT Ling et al. (2025) utilizes a VLM to generate a static 3D cost map of the environment, assigning higher costs to fragile objects to enable a planner like RRT* to find paths with "acceptable contact". Similarly, VLMPC Zhao et al. (2024) embeds a VLM within a Model Predictive Control (MPC) Garcia et al. (1989) loop, where it provides perceptual guidance by identifying sub-goals and sampling candidate action sequences. CoRAL significantly advances this paradigm by elevating the role of the LLM from a perceptual guide to a high-level strategist. Instead of merely identifying objects or goals, our LLM formulates the structure of the Model Predictive Path Integral (MPPI) Williams et al. (2017) controller cost function itself and proposes symbolic contact strategies. This approach grounds abstract, commonsense reasoning directly into the mathematical formulation of the optimal control problem, enabling a more profound and explainable link between high-level intent and low-level dynamic execution.

**Tackling Contact-Rich Manipulation** Contact-rich manipulation remains a grand challenge, as it requires nuanced force control and physical understanding beyond simple trajectory generation. One major line of work addresses this by augmenting perception with physical sensors, such as in ForceVLA Yu et al. (2025), TLA Hao et al. (2025) and VLA-Touch Bi et al. (2025), which explicitly integrate force or tactile data into the policy's input stream. Further research in this direction, such as Reactive Diffusion Policy (RDP) Xue et al. (2025) and FACTR Liu et al. (2025), proposes specialized architectures and training curricula to more effectively fuse this real-time feedback into a learned policy. While effective, this hardware-centric approach risks creating a new data bottleneck, as it requires large-scale, specialized multimodal demonstration datasets that are difficult to collect Firoozi et al. (2025). CoRAL tackles this problem from a different angle. Although it also leverages real-time force feedback for its reactive controller, it critically eliminates the need for prior demonstration datasets containing such data. Instead, our framework uses the LLM to formulate a high-level strategy and cost function that explicitly reasons about interaction forces, which the MPPI controller then robustly executes by adapting to live sensor data online. This neuro-symbolic approach combines the benefits of physical sensing with the zero-shot reasoning of foundation models, thereby avoiding the imitation learning bottleneck while still achieving precise, force-aware control.

## 3 METHODOLOGY

CoRAL is a neuro-symbolic framework designed for zero-shot, contact-rich manipulation. It strategically decouples high-level reasoning from low-level control by integrating a vision pipeline that continuously tracks object poses and enriches the world model with physical parameters estimated by the VLM, a LLM acting in two distinct roles (Task Formulation and Online Adaptation), a Memory Unit for experience retrieval, and a Model Predictive Path Integral controller (MPPI) for reactive execution. The overall architecture, which features nested feedback loops for rapid and robust adaptation, is illustrated in Figure 2. Below, we detail each component of this architecture.

### 3.1 ENVIRONMENT PERCEPTION AND WORLD MODEL INITIALIZATION

The first step is to translate raw visual, textual, and geometric inputs into a structured, physics-aware world model. Our perception pipeline achieves this through a two-stage process that first establishes the geometric state of the scene and then identifies it with physical properties. The process involves two core steps:

1. **Pose Estimation and Tracking:** We employ **FoundationPose** Wen et al. (2024), a state-of-the-art pose estimation model, to determine and continuously track the 6-DoF poses of

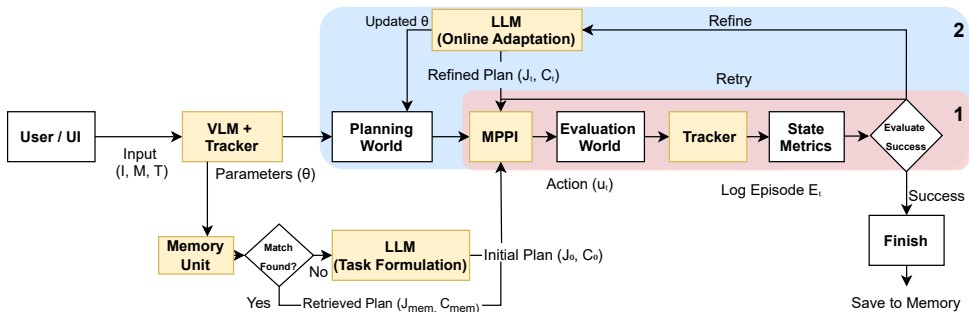

Figure 2: The overall architecture of the **CoRAL** framework. Given an input image $I$, object models $M$ and task description $T$, the vision module extracts world parameters $\theta$. If the Memory Unit finds a similar successful experience, its retrieved plan $(J_{\text{mem}}, C_{\text{mem}})$ is used to guide the MPPI controller. Otherwise, the LLM (Task Formulation) module generates an initial plan $(J_0, C_0)$. The system then enters the main execution cycle, which is governed by two nested feedback loops labeled (1) and (2). **(1) The Inner Loop** is a high-frequency re-planning cycle. At each step, the MPPI, guided by the current plan, generates an action $u_t$ based on the latest 'State Metrics'. This loop ('Retry') continues until the task succeeds or a refinement is needed. **(2) The Outer Loop** is a low-frequency, high-level adaptation cycle. If the inner loop fails persistently, the 'Refine' path is taken, where the LLM (Online Adaptation) updates both the world model parameters ($\theta$) for the 'Planning World' and the strategic 'Refined Plan $(J_t, C_t)$' for the MPPI. Successful episodes are stored back into the Memory Unit.

     all interactable objects. This model takes the RGB-D camera images $I$, and the known 3D geometric models of the objects, $M$, as input. The output is a real-time stream of estimated pose data for each object in the scene.

2. **Physical Parameter Estimation:** The pose data, along with the visual input and the language-based task description $T$, is then fed into a multimodal foundation model (GPT-4o), which acts as our VLM. The VLM's role is to leverage its extensive world knowledge to infer the unobservable physical properties of the objects based on their appearance and the context provided by the task. It estimates crucial attributes like *mass* and *friction coefficients*, which are vital for accurate physical simulation.

The combined output of this perception pipeline is a structured set of world parameters, $\theta$. For each object, $\theta$ contains its semantic label (derived from the input 3D model), its continuously tracked pose from FoundationPose, and its estimated physical attributes from the VLM. These parameters are crucial as they are used to construct and continuously update the internal `Planning World` that the MPPI planner operates on.

## 3.2 LLM-DRIVEN TASK FORMULATION AND MEMORY RETRIEVAL

Once the world is perceived, the system formulates a concrete plan. This is handled by the 'LLM (Task Formulation)' module, which can generate a plan from scratch or leverage past experiences from the 'Memory Unit'.

**Memory Retrieval:** Before invoking the LLM, the system queries the 'Memory Unit' with the current world parameters $\theta$ and task $T$. Our memory module is based on Retrieval-Augmented Generation (RAG), storing successful "experience episodes" indexed by task definitions and environmental parameters. Instead of relying on predefined similarity metrics, the LLM embeds the current task into a latent semantic space to retrieve the most relevant past experience:

$$(J_{\text{mem}}, C_{\text{mem}}) = \text{RAG}_{\text{Retrieve}}(T, \theta) \tag{1}$$

where $J_{\text{mem}}$ denotes the final cost function that led to a successful episode, and $C_{\text{mem}}$ denotes the corresponding contact strategy, which also resulted in a successful completion of the task. If a sufficiently similar and successful past experience is found, its stored plan $(J_{\text{mem}}, C_{\text{mem}})$ is retrieved and

used as the initial plan, bypassing the computationally expensive initial LLM call and accelerating performance.

**Plan Generation from Scratch:** If no suitable memory is found, the 'LLM (Task Formulation)' module is invoked. It acts as a high-level strategist, translating the task $T$ and world parameters $\theta$ into a formal optimization problem. Its output is an initial plan tuple $(J_0, C_0)$, where:

- **Initial MPPI Cost Function ($J_0$):** The LLM generates the mathematical structure and relative weights of a cost function. Specifically, for a given task, the LLM provides a structured cost functional, for instance:

$$J_0(\mathbf{x}_{0:H}, \mathbf{u}_{0:H-1}) = \sum_{t=0}^{H-1} \Big[ w_d \left\| \mathbf{p}_{\text{target}} - \mathbf{p}_{\text{obj}}(t) \right\|^2 \\ + w_c \, \mathbb{I}\{\text{no contact at } t\} \\ + w_u \left\| \mathbf{u}_t \right\|^2 \Big] \quad (2)$$

Here, the weights $w_d, w_c, w_u$ are determined by the LLM based on the task description (e.g., for a pushing task, $w_c$ would be high). $\mathbf{p}_{\text{obj}}(t)$ is the object's tracked position at time $t$, and $\mathbb{I}\{\cdot\}$ is an indicator function penalizing the absence of contact. This expression is only an illustrative example: in general, the LLM is free to introduce any cost terms constructible from the available state, pose, and action variables, and is not restricted to a fixed finite set of cost terms.

- **Initial Contact Strategy ($C_0$):** The LLM proposes promising surfaces for making contact to guide the planner's exploration. It specifies a set of focused surface regions $\{R_j\}$, from which we generate candidate contact points as:

$$C_0 = \bigcup_{j=1}^{M} \Big\{ c_j + e_j \big( \cos\phi \, \mathbf{t}_{1,j} + \sin\phi \, \mathbf{t}_{2,j} \big) \\ \Big| \, \phi = \tfrac{2\pi k}{N_j}, \; k = 0, \ldots, N_j - 1 \Big\} \quad (3)$$

where for each region $j$, $c_j$ is the center, $e_j$ is the radius, and $\{\mathbf{t}_{1,j}, \mathbf{t}_{2,j}\}$ are tangent vectors. This biases sampling towards strategically advantageous regions.

This generated strategy is then used within the MPPI sampling process by biasing the initial control perturbations to explore actions that bring the end-effector closer to the LLM-proposed contact regions, thereby significantly pruning the search space.

### 3.3 REACTIVE PLANNING AND EXECUTION (THE INNER LOOP)

The core of our system is a high-frequency, reactive execution cycle governed by the MPPI controller. This corresponds to the Inner Loop (1) in Figure 2.

**MPPI Formulation:** The MPPI controller solves a stochastic optimal control problem at each timestep. Given a state-transition model $x_{t+1} = f(x_t, u_t) + \epsilon_t$, where $x_t$ is the system state, $u_t$ is the control input, and $\epsilon_t$ is system noise, the objective is to find a sequence of control inputs $U = \{u_0, \ldots, u_{H-1}\}$ that minimizes the expected total cost:

$$U^* = \arg\min_U \mathbb{E} \left[ \phi(x_H) + \sum_{t=0}^{H-1} q(x_t, u_t) \right] \quad (4)$$

where $\phi(x_H)$ is a terminal state cost and $q(x_t, u_t)$ is the running cost at each step. The LLM-generated cost function, $J_0$ (from Eq. 2), directly defines the terms used in this optimization. Specifically, the running cost $q(x_t, u_t)$ is the expression inside the summation of $J_0$:

$$q(x_t, u_t) = w_d \left\| \mathbf{p}_{\text{target}} - \mathbf{p}_{\text{obj}}(t) \right\|^2 + w_c \, \mathbb{I}\{\text{no contact at } t\} + w_u \left\| \mathbf{u}_t \right\|^2 \quad (5)$$

MPPI approximates this optimization by:

1. Sampling $K$ control sequence perturbations $\delta U_k \sim \mathcal{N}(0, \Sigma)$ from a Gaussian distribution.

2. Creating $K$ rollout trajectories by applying the perturbed control sequences $V_k = U_{prev} + \delta U_k$ in the 'Planning World'.

3. Calculating the total cost $S(V_k)$ for each of the $K$ trajectories.

4. Computing an exponentially weighted average of the perturbations to update the control sequence:

$$U_{new} = U_{prev} + \sum_{k=1}^{K} w_k \delta U_k \quad \text{where} \quad w_k = \frac{\exp\left(-\frac{1}{\lambda}S(V_k)\right)}{\sum_{j=1}^{K}\exp\left(-\frac{1}{\lambda}S(V_j)\right)} \quad (6)$$

Following the receding horizon principle, only the first action, $u_0$, of the newly optimized sequence $U_{new}$ is executed.

**Reactive Control Augmentation:** To achieve robustness against the inherent sim-to-real gap, we augment the nominal planned action with a real-time feedback term. The final control command $\nu_t$ sent to the robot is:

$$\nu_t = u_t + K_f \cdot (x_{\text{des}} - x_{\text{measured}}) \quad (7)$$

where $u_t$ is the action computed by MPPI, the error term is calculated from real-time sensors (e.g., force/torque, proprioception), and $K_f$ is a feedback gain matrix. This 'Retry' loop continues at a high frequency, constantly re-planning and correcting based on physical feedback.

### 3.4 ONLINE ADAPTATION VIA LLM-DRIVEN REFINEMENT (THE OUTER LOOP)

If the inner loop fails to make progress after a predefined number of attempts, a hyperparameter we denote as $N_{\text{retry}}$, the system triggers the low-frequency Outer Loop (2). This invokes the 'LLM (Online Adaptation)' module, which acts as a diagnostician and re-strategist.

The input to this module is the logged episode data $E_t$, which contains the history of states, actions, the contact strategies and cost functions that were used, and the estimated physical parameters that led to the failure. By analyzing this rich context, the LLM performs two critical functions:

1. **World Model Correction:** The LLM can refine the initial physical parameter estimates. For example, if the robot pushes an object but the object moves less than predicted, the LLM can infer that its initial estimate of the object's mass was too low and output an 'Updated $\theta$'.

2. **Strategy Refinement:** The LLM can also alter the plan itself. It might change the weights of the cost function (e.g., prioritizing force control over position accuracy) or propose an entirely new contact strategy. This results in a 'Refined Plan $(J_t, C_t)$'.

This refined world model and plan are then fed back into the inner loop, allowing the system to learn from its failures and adapt its entire approach within a single task execution.

## 4 EXPERIMENTS

We conducted a series of experiments in a simulated environment to rigorously evaluate the performance of CoRAL. Our evaluation is designed to answer three key research questions: **(RQ1)** How does CoRAL perform on complex, contact-rich manipulation tasks in a zero-shot setting compared to state-of-the-art baselines? **(RQ2)** How critical is each core component of our neuro-symbolic architecture—specifically the vision/language model role separation, the online refinement loop, and the memory unit—to the overall success? **(RQ3)** Can the system demonstrate robustness and adaptability by reasoning about and recovering from failures?

**Simulation Environment:** All experiments were conducted in the evaluation world, implemented using ROBOSUITE library Zhu et al. (2020), which is based on the MUJOCO physics engine Todorov et al. (2012). The robot is a simulated 7-DoF Franka Emika Panda arm with a parallel-jaw gripper. Sensory inputs include RGB-D images from a fixed camera, proprioceptive feedback, and force/-torque data provided by the physics engine, which are essential for our reactive control. In addition

to our custom environments, two benchmark tasks from the LIBERO suite Liu et al. (2023) were also incorporated for evaluation.

**Implementation Details:** The VLM and LLM modules were implemented using the GPT-4o API. The MPPI controller was integrated on top of the ROBOSUITE/MUJOCO environment. To improve computational efficiency, our MPPI implementation parallelizes the rollout of $K = 200$ trajectories over a planning horizon of $H = 50$ steps at each control cycle on the CPU. The key hyperparameters for the controller are the temperature $\lambda = 0.1$ and the outer-loop trigger threshold $N_{retry} = 15$ persistent failures. The entire framework was run on a desktop computer with an Intel Core i9-13900K CPU, 64 GB of RAM, and a single NVIDIA RTX 4060 Ti GPU.

**Tasks and Evaluation Metrics:** We evaluated our framework on six challenging, contact-rich manipulation tasks, shown in Fig. 3, designed to be difficult for purely vision-based, collision-avoidant planners. Each task was performed 10 times with randomized initial object poses, object masses, surface friction coefficients and the object dimension for the box object. The tasks are as follows: **T1: Push and Pick Cutting Board**, a multi-stage task testing pushing and reasoning about object parts and pose for grasping; **T2: Pick Box & T3: Pick and Place in Clutter**, a standard pick-and-place task to establish a baseline; **T4: Push with Constant Force**, testing the reactive controller's ability to manage force feedback; **T5: Flip Box**, a dynamically complex maneuver and; **T6: Flip with Wall**, requiring multi-contact reasoning to use the wall as a fixture; **Evaluation Metrics:** We use two primary metrics: **Success Rate** (binary measure across 10 trials) and **Average Completion Time** in seconds for successful trials.

| Push & Pick Cutting Board | Pick & Place Box | Pick & Place in Clutter | Push with Constant Force | Flip the Box | Flip with Wall |

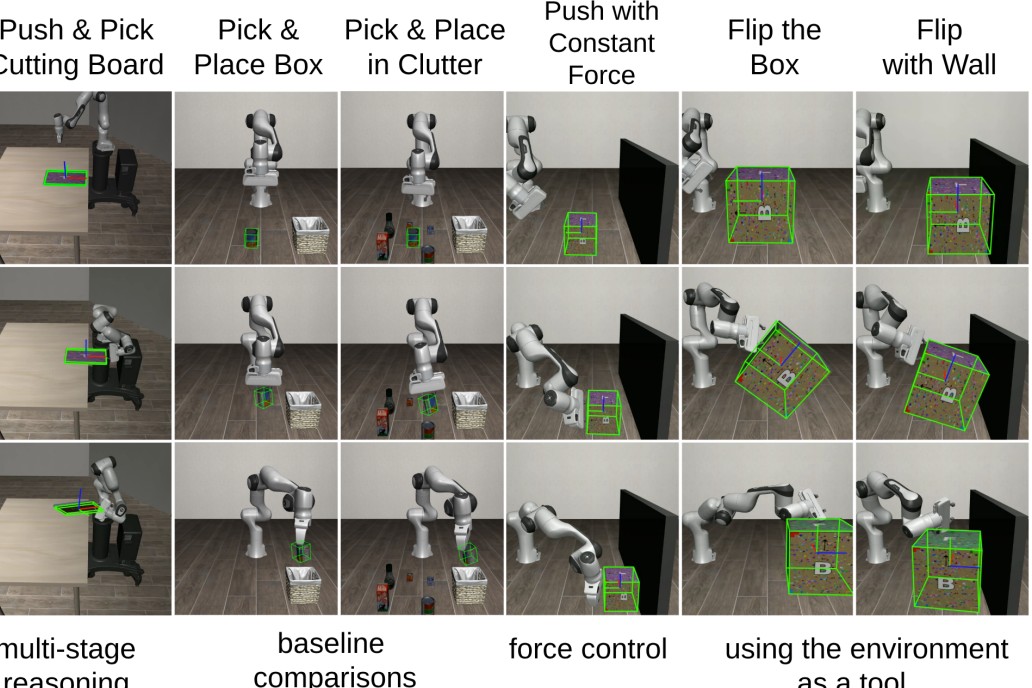

| multi-stage reasoning | baseline comparisons | force control | using the environment as a tool |

Figure 3: CoRAL on six different tasks with the tracked pose overlay of the object of interest.

**Comparative Baselines:** We compare CoRAL against two state-of-the-art methods and four internal ablations. The **State-of-the-Art Baselines** are: **OpenVLA-OFT** Kim et al. (2025a) and $\pi_{0.5}$ Black et al. (2025), two leading end-to-end VLA models. For each model, we rely on the officially released LIBERO-OBJECT checkpoint for pick-and-place tasks and the LIBERO-GOAL checkpoint for all other tasks. This setup tests CoRAL's zero-shot capabilities against powerful policies. In addition, we include two **Human Expert-Designed Cost** baselines. In the *single-stage* variant, an expert manually designs a single MPPI running-cost for each task. In the *FSM* variant, the expert is allowed to construct an explicit finite-state machine with phase-specific costs (e.g.,

Table 1: Comprehensive comparison against the state-of-the-art baselines and ablation study of our method variants across all tasks. Performance is measured by success rate (x/10 trials) and average completion time in seconds for successful trials.

| Method | T1: Push+Pick Board | | T2: Pick+Place Box | | T3: Pick+Place Clutter | | T4: Push Const. Force | | T5: Flip Box | | T6: Flip w/ Wall | |
|---|---|---|---|---|---|---|---|---|---|---|---|---|
| | Success | Time (s) | Success | Time (s) | Success | Time (s) | Success | Time (s) | Success | Time (s) | Success | Time (s) |
| *State-of-the-Art Baseline* | | | | | | | | | | | | |
| OpenVLA-OFT Kim et al. (2025a) | 0/10 | - | 10/10 | **5** | 9/10 | **7** | 0/10 | - | 1/10 | **9** | 0/10 | - |
| $\pi_0.5$ Black et al. (2025) | 0/10 | - | 10/10 | 12 | 8/10 | 13 | 0/10 | - | 3/10 | 21 | 0/10 | - |
| *Human Expert-Designed Cost Baselines* | | | | | | | | | | | | |
| Expert (hand-designed cost, single-stage) | 0/10 | - | 10/10 | 38 | 10/10 | 40 | 9/10 | 32 | 9/10 | 47 | 3/10 | 79 |
| Expert (hand-designed costs, FSM) | 8/10 | 154 | 10/10 | 40 | 10/10 | 44 | 10/10 | 48 | 10/10 | 58 | 9/10 | 95 |
| *Our Method (Ablation Study)* | | | | | | | | | | | | |
| **CoRAL (Ours, with Memory)** | **4/10** | **162** | **10/10** | 45 | **10/10** | 49 | **9/10** | **52** | **9/10** | 63 | **7/10** | **108** |
| CoRAL (w/o Memory) | 2/10 | 212 | 10/10 | 54 | 9/10 | 61 | 9/10 | 109 | 7/10 | 98 | 5/10 | 164 |
| CoRAL (w/o Refinement) | 0/10 | – | 10/10 | 42 | 3/10 | 36 | 6/10 | 34 | 4/10 | 51 | 2/10 | 92 |
| CoRAL (Unified VLM) | 0/10 | – | 2/10 | 21 | 0/10 | – | 1/10 | 35 | 0/10 | – | 0/10 | – |
| CoRAL (w/o Pose Tracking) | 0/10 | – | 0/10 | – | 0/10 | – | 0/10 | – | 0/10 | – | 0/10 | – |

push–then–pick or push–then–flip). In both cases, the cost functions are tuned in a separate design environment and then evaluated *as-is* in our randomized test environment, providing an upper bound on what carefully engineered, task-specific objectives can achieve. Our **Ablation Baselines** are: **CoRAL (w/o Pose Tracking)**, which removes FoundationPose and relies on the VLM to estimate object poses, testing the criticality of a dedicated pose estimator; **CoRAL (w/o Memory)**, which removes the experience retrieval mechanism; **CoRAL (w/o Refinement)**, which disables the online adaptation loop; and **CoRAL (Unified VLM)**, which uses a single multimodal prompt for both perception and planning to test the importance of separating VLM/LLM roles.

## 4.1 RESULTS AND ANALYSIS

Table 1 presents a comprehensive overview of our experimental findings.

### 4.1.1 STATE-OF-THE-ART COMPARISON (RQ1)

CoRAL significantly outperforms both state-of-the-art baselines, OpenVLA-OFT and $\pi_{0.5}$, particularly in tasks requiring sophisticated physical reasoning (T1, T4, T5, T6). While both baselines perform well on the simpler pick-and-place tasks (T2, T3), their performance degrades sharply on the more complex, contact-rich scenarios. This is a critical finding, as both models were fine-tuned on the LIBERO benchmark, which should theoretically demonstrate some generalization for manipulation tasks in similar environments. However, our results indicate that even fine-tuning an end-to-end policy is insufficient for scenarios that demand explicit physical modeling and reasoning about forces or multi-contact strategies. These policies fail to generate the non-obvious maneuvers required for tasks like the wall-flip (T6) or maintaining steady force for pushing (T4). In contrast, our framework excels by allowing the LLM to directly formulate a cost function that optimizes for these physical interaction dynamics, enabling robust zero-shot execution without any task-specific fine-tuning.

### 4.1.2 COMPARISON TO HUMAN-DESIGNED COST FUNCTIONS

The two human baselines approximate an upper bound from carefully engineered,task-specific objectives. As expected, the *Expert (FSM)* variant achieves the strongest overall performance, and the single-stage expert design remains competitive, particularly on simpler tasks such as T2–T4, where CoRAL largely matches but does not surpass its success rate and speed (Table 1). On more sequential and contact-heavy tasks (T1, T5, T6), CoRAL narrows the gap to the expert, achieving higher success rates than the single-stage baseline while remaining below the FSM upper bound. This shows that our LLM-based controller can recover much of the structure of expert-designed costs automatically, substantially reducing manual tuning effort while approaching expert-level performance on the hardest tasks.

### 4.1.3 ABLATION STUDY ANALYSIS (RQ2)

Our ablation studies clearly demonstrate the necessity of each component in our architecture.

**The Synergy of Separated VLM/LLM Roles:** The *CoRAL (Unified VLM)* variant, which tasked a single VLM with both perception and planning, failed on nearly all complex tasks. This starkly illustrates our core hypothesis: separating the role of a VLM for perception from a dedicated LLM for strategy formulation is crucial for robust performance. The specialized modules provide more reliable and structured outputs for the planner.

**The Importance of Online Refinement:** The *w/o Refinement* variant showed a dramatic performance drop in multi-stage tasks like T1 (Push and Pick Board), with success falling from 4/10 to 0/10. In this task, the initial plan often failed because the VLM's initial friction estimate was slightly off, causing the board to slip during the pick. The full CoRAL framework, however, used the outer loop for the LLM to diagnose this from the physical outcome, refine the friction parameter in its world model, and successfully complete the task. This shows the system's ability to learn from failure.

**The Benefit of Experience Reuse:** The full framework *with Memory* consistently achieved the highest success rates and fastest completion times. For instance, in T1 and T3, memory boosted the success rate from 2/10 to 4/10 and 9/10 to 10/10, respectively. By retrieving a successful "push-to-edge" strategy from a past experience, the system provided the planner with a superior initialization, accelerating convergence and leading to more robust solutions.

**The Criticality of a Dedicated Pose Estimator:** The *w/o Pose Tracking* ablation, which removed FoundationPose and relied solely on the VLM for pose estimation, resulted in a catastrophic failure across all tasks (0/10 success). The VLM, while powerful for semantic understanding, is ill-suited for the precision required by 6-DoF pose tracking through dynamic interactions. It frequently produced trivial or physically impossible pose estimations ("hallucinations") that rendered the planner's output useless. This result provides conclusive evidence that a dedicated, high-fidelity pose estimator is not merely beneficial but a critical and non-negotiable component of our architecture, serving as the geometric foundation upon which all subsequent physical reasoning is built.

### 4.1.4 Robustness Analysis (RQ3)

**Analysis of LLM-Guided Contact Strategy:** To isolate the contribution of the LLM's initial contact strategy ($C_0$), we conducted a targeted ablation on the challenging "Flip with Wall" task (T6). We compared the performance of our full framework against a variant where the LLM only provided the cost function ($J_0$), forcing the MPPI planner to rely on uninformed, random sampling to find useful contact points.

The guided trajectory (With Strategy, green) is direct and purposeful, immediately moving the end-effector to the correct corner of the box to initiate the flip (Figure 5 in Appendix). In contrast, the unguided trajectory (Without Strategy, red) is chaotic and inefficient, exploring large, irrelevant portions of the workspace. The planning cost for the unguided agent remains high and erratic, indicating a constant struggle to find a viable plan. This visual difference is confirmed by the quantitative results: the guided approach was **83.9% faster** (32 vs. 199 steps) and the end-effector traveled a **63.9% shorter path** (1.33 m vs. 3.69 m). This analysis provides clear evidence that the LLM's symbolic contact strategy is critical for transforming a computationally intractable, long-horizon contact problem into a solvable one by intelligently pruning the vast action search space.

**Robustness of Online Parameter Adaptation:** Beyond strategy and cost function refinement, CoRAL's 'Online Adaptation' module, driven by the LLM, exhibits a crucial ability to correct the agent's internal world model online. To demonstrate this, we intentionally initialized the *Evaluation World* with a severely overestimated mass (2.0 kg vs. a ground truth of 0.1 kg) and friction coefficient (0.9 vs. 0.5) for the cutting board. These initial biases represent a severe sim-to-real gap or a VLM hallucination.

Figure 4 vividly illustrates the adaptation process. The LLM's 'Online Adaptation' module, when triggered by persistent failures in the inner loop (e.g., the board not moving as expected despite high pushing force), diagnosed the discrepancy. Through an iterative refinement process, it progressively adjusted its estimated mass and friction parameters. As shown in the graph, after several adaptation cycles, the agent's belief about both mass and friction converged remarkably close to their true values. This online correction of physical parameters is fundamental to the framework's robustness, allowing it to overcome initial environmental mischaracterizations and successfully execute contact-rich tasks that would otherwise fail due to a misaligned internal world model. This capability is a

cornerstone for deploying robots in unknown environments where accurate a priori physical models are often unavailable.

**Sequential Reasoning and Experience Reuse in the Cutting Board Task** The "Push and Pick Cutting Board" task (T1) is specifically designed to test the framework's capabilities in long-horizon, sequential manipulation. The task's difficulty lies in its two distinct phases: a stable push across a surface until a portion of the board is exposed for grasping followed by a precise grasp of the board. As evidenced by our results in Table 1, this sequential challenge

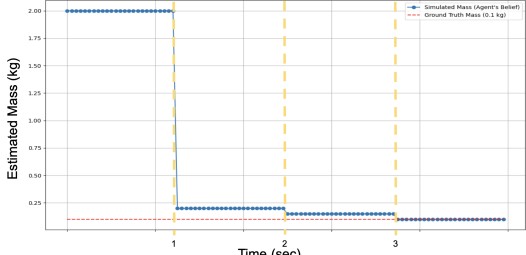

Figure 4: Object mass correction.

highlights the importance of two core components of our architecture: online adaptation and experience reuse.

First, the long-horizon nature of the task means that small errors in the initial world model can accumulate and lead to failure in the later stages. This is clearly demonstrated by the *w/o Refinement* ablation, which failed entirely on this task (0/10 success rate). While its initial plan was often sufficient for the pushing phase, slight inaccuracies in the estimated friction parameter caused the board to end up in an unexpected final pose, leading to a failed grasp. Our full model, however, leverages the outer loop to learn from the outcome of the push, allowing the 'LLM (Online Adaptation)' to refine its friction estimate and update the plan for the subsequent pick, thereby enabling success.

Second, this task powerfully illustrates the benefit of the 'Memory Unit'. The performance of our full model without memory was respectable (2/10), but the inclusion of the memory module boosted the success rate significantly to 4/10. This shows that after just a single successful completion, the system can store the entire successful interaction context (the refined parameters, cost function, and contact strategy). When faced with a similar task configuration, it retrieves this proven strategy, providing the MPPI planner with a superior initialization that leads to more robust and efficient execution. This result demonstrates that CoRAL can learn from its experiences, and it highlights a clear path towards few-shot performance improvements where the system becomes more adept as it gathers successful episodes.

**Explainability and Automated Failure Recovery** A key advantage of our neuro-symbolic design is its inherent explainability, particularly during failure recovery. Unlike opaque end-to-end models, CoRAL can articulate *why* it failed and *what* it is doing to correct its plan. We demonstrate this with a scenario where the "Flip with Wall" task persistently fails, triggering the Outer Loop.

Instead of just outputting a new set of parameters, the 'LLM (Online Adaptation)' module provides a full natural language diagnosis of the failure and a detailed log of the corrective actions it is taking. The LLM provided a correct natural language diagnosis of a poorly weighted cost function and proceeded to adjust the specific weights to remedy the failure (Appendix **??**). This capability to "think out loud" is a critical feature for building trust and diagnosing failures in complex robotic systems, providing a level of transparency that is simply not possible with black-box policies.

## 5 LIMITATIONS & CONCLUSION

In this paper, we introduced **CoRAL**, a novel framework that addresses the challenges of zero-shot, contact-rich manipulation. Our approach departs from conventional end-to-end paradigms by integrating foundation models with a reactive controller. Experiments on challenging tasks demonstrate that this modular, synergistic design enables the system to adapt to unseen scenarios without prior demonstrations, significantly enhancing both performance and explainability over monolithic approaches. While promising, the framework's performance is currently contingent on the fidelity of the vision-based world model and is subject to computational latency from its sequential pipeline. These limitations and future research directions are discussed in detail in Appendix A.3.2. We believe this hybrid paradigm—coupling large-scale, pre-trained knowledge with rigorous real-time control—is a promising direction for creating more capable and physically intelligent robotic agents.

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

# A APPENDIX

## A.1 QUALITATIVE COMPARISON WITH STATE-OF-THE-ART VLA MANIPULATION FRAMEWORKS

Table 2: Comparison with State-of-the-Art VLA Manipulation Frameworks

| Framework | Primary Modality | Planning & Control Strategy | Reasoning Mechanism | Data Requirement |
|---|---|---|---|---|
| OpenVLA Kim et al. (2025b) | Vision, Language | End-to-End Learned Policy (Action Token Prediction) | Implicit (in VLM backbone) | Large-scale Imitation Learning Demos |
| $\pi_0$ (pi-zero) Black et al. (2024) | Vision, Language | End-to-End Learned Policy (Flow Matching) | Implicit (in VLM backbone) | Large-scale Imitation Learning Demos |
| ForceVLA Yu et al. (2025), TLA Hao et al. (2025) | Vision, Language, **Tactile/Force** | End-to-End Learned Policy | Implicit (in network weights) | Large-scale Tactile/Force Demos |
| VLA-Touch Bi et al. (2025) | Vision, Language, **Tactile** | VLA Policy + Tactile-based Refinement Controller | Explicit VLM Planning + Semantic Tactile Feedback | Leverages pretrained models; no VLA fine-tuning |
| ThinkAct Huang et al. (2025), ECoT Zawalski et al. (2024) | Vision, Language | End-to-End Learned Policy | **Explicit LLM Reasoning** (Chain-of-Thought) | Large-scale Imitation Learning Demos |
| OneTwoVLA Lin et al. (2025) | Vision, Language | Unified Policy (Adaptive Acting & Reasoning) | **Explicit LLM Reasoning** (System 2) | Imitation Demos + Synthetic Reasoning Data |
| MolmoAct Lee et al. (2025) | Vision, Language | Multi-stage Pipeline (Perception, Spatial Plan, Action) | **Explicit Spatial Reasoning** (Trajectory Traces) | Large-scale Imitation Learning Demos |
| IMPACT Ling et al. (2025) | Vision, Language | VLM-based Static Cost Map + RRT* | Implicit (semantic object labeling) | N/A (Planner-based) |
| VLMPC Zhao et al. (2024) | Vision, Language | VLM-guided Model Predictive Control (MPC) | **Explicit VLM Reasoning** (for cost & sampling) | N/A (Planner-based) |
| **CoRAL (Ours)** | Vision, Language, **Tactile/Force** | **LLM-guided MPPI + Reactive Control** | **Explicit LLM Reasoning** (Strategy Formulation) | **Zero-Shot (No Demos)** |

As the comparative analysis in Table 2 illustrates, the field of robotic manipulation has historically involved a trade-off. End-to-end VLA models achieve impressive behaviors but are fundamentally constrained by large-scale demonstration datasets, while traditional planner-based systems are zero-shot but often lack high-level semantic reasoning. CoRAL is designed to synthesize the strengths of these disparate paradigms. To the best of our knowledge, CoRAL is the first framework to simultaneously integrate **explicit LLM-driven strategy formulation** with a **dynamic, reactive controller** that leverages real-time **tactile and force feedback**, all while operating in a **zero-shot** manner that completely eliminates the need for prior demonstration data.

## A.2 PRELIMINARIES

In this section, we introduce the Model Predictive Path Integral (MPPI) controller, which forms the core of our methodology, and provide a formal problem formulation for the contact-rich manipulation tasks we address.

### A.2.1 MODEL PREDICTIVE PATH INTEGRAL (MPPI)

Model Predictive Path Integral (MPPI) Williams et al. (2017) is a sampling-based Model Predictive Control (MPC) Garcia et al. (1989) algorithm designed to solve stochastic optimal control problems. It is particularly effective for systems with nonlinear and complex dynamics. MPPI operates by simulating thousands of potential control sequences in parallel from the current state to determine the optimal subsequent control input.

Consider a system with discrete-time stochastic dynamics described by $\mathbf{x}_{t+1} = f(\mathbf{x}_t, \mathbf{u}_t) + \boldsymbol{\epsilon}_t$, where $\mathbf{x}_t$ is the state of the system, $\mathbf{u}_t$ is the control input, and $\boldsymbol{\epsilon}_t$ represents system noise. The objective of MPPI is to find a control sequence $\mathbf{U} = \{\mathbf{u}_0, \mathbf{u}_1, \dots, \mathbf{u}_{H-1}\}$ that minimizes an expected cost function:

$$J(\mathbf{U}) = \mathbb{E}\left[\phi(\mathbf{x}_H) + \sum_{t=0}^{H-1} q(\mathbf{x}_t, \mathbf{u}_t)\right] \tag{8}$$

Here, $\phi(\mathbf{x}_H)$ is the terminal cost, and $q(\mathbf{x}_t, \mathbf{u}_t)$ is the running (or stage) cost, which typically includes terms for tracking error and control effort. $H$ is the planning horizon.

At each time step, MPPI samples $K$ candidate control sequences by adding random noise perturbations to a nominal sequence. These are rolled out in simulation to get trajectories, and the total cost $S_k$ for each is computed. The optimal control is then calculated via an exponential weighting of these costs. This process is repeated at each time step following the receding horizon principle. In our approach, the symbolic reasoning provided by the LLM forms the initial structure for the running cost function $q(\cdot)$.

### A.2.2 PROBLEM FORMULATION

In this work, we address long-horizon, contact-rich manipulation tasks specified by visual and linguistic commands. Our goal is to develop a robotic system capable of interacting with objects of unknown physical properties (e.g., mass, friction) and generalizing to new scenarios in a zero-shot manner.

We formulate the problem as a POMDP. The state of the system, $\mathbf{x}_t \in \mathcal{X}$, includes the robot's state $\mathbf{x}_r(t)$ and the states of environmental objects $\mathbf{x}_o(t)$. The action space, $\mathcal{U}$, consists of continuous control commands applicable to the robot's end-effector.

At the beginning of each task, the system receives an RGB-D image $I$, a natural language instruction $T$, and the corresponding 3D object models $M$. The system's objective is not to learn a fixed policy, but rather to compute, at each step, a control action $u_t$ that leads to a sequence of actions $\mathbf{U} = \{\mathbf{u}_0, \ldots, \mathbf{u}_{H-1}\}$ that successfully completes the task.

The core challenge is to bridge the gap from high-level, multimodal inputs $(I, M, T)$ to these low-level, continuous control actions. Our approach decomposes this problem into two stages:

1. **Strategy Formulation:** We use the vision module and LLM to translate the multimodal inputs $(I, M, T)$ into an initial cost function $J_0$ and contact candidates $C_0$ for the planner.

2. **Online Planning and Control:** We employ the MPPI planner to compute the optimal action sequence online, guided by the LLM's strategy and adapted using real-time sensory feedback.

This formulation accurately frames our system as an online planner that reasons and computes actions on the fly, rather than an agent executing a pre-learned, static policy.

## A.3    DISCUSSION & LIMITATIONS

Our work introduces CoRAL, a framework that represents a deliberate architectural shift away from the prevailing end-to-end paradigm for Vision-Language-Action (VLA) models. By decoupling high-level reasoning from low-level, physics-based planning and control, we address several fundamental challenges in contact-rich manipulation, particularly regarding explainability, data efficiency, and physical grounding.

### A.3.1    DISCUSSION

Our experimental results validate the core hypotheses of CoRAL, demonstrating that a modular, neuro-symbolic architecture can overcome the fundamental limitations of end-to-end models in complex, contact-rich manipulation. We discuss the key implications of our findings below.

**The Synergy of Grounded Reasoning and Reactive Control:** Our experiments reveal a clear synergy between high-level reasoning and low-level reactive control. The performance gap between CoRAL and end-to-end baselines like OpenVLA, especially in tasks requiring non-trivial strategies (e.g., T6: Flip with Wall), highlights the brittleness of purely imitative policies. These models fail because their training data lacks examples of using the environment as a tool. In contrast, CoRAL's success stems from its ability to reason: the LLM formulates an explicit optimization problem ($J_0$) that defines the task's success conditions, while the MPPI controller finds a physically plausible solution. This makes the system's intent transparent and grounds abstract reasoning in a formal control framework, a more robust approach than conditioning a black-box policy as is done in works like ECoT Zawalski et al. (2024).

**Online Adaptation as an Alternative to Large-Scale Tactile Datasets:** A significant implication of our work is a path away from the data-hungry paradigm of modern robotics. State-of-the-art methods like ForceVLA Yu et al. (2025) achieve impressive results by incorporating tactile feedback, but this requires creating massive, specialized demonstration datasets. Our results demonstrate a more data-efficient alternative. CoRAL also uses real-time force feedback, but its role is redefined: it serves as a signal for *online adaptation*, not offline imitation. The success of our outer feedback loop, particularly in tasks where initial parameter estimates were deliberately inaccurate, proves that the LLM can diagnose physical failures and refine its world model on the fly. This ability to learn from direct interaction significantly lowers the barrier to entry for creating sophisticated, contact-aware robots without relying on pre-collected, large-scale tactile data.

**Zero-Shot Planning and the Path to Lifelong Learning:** Perhaps the most significant result is CoRAL's ability to perform zero-shot planning for novel tasks. This capability stems from its core design: instead of learning *how to act*, it leverages the pre-existing knowledge of foundation models to reason about *how to plan*. By generating a cost function from a single image and a language com-

mand, the system dynamically tackles new objectives. The consistent performance boost provided by the 'Memory Unit' in our ablation studies further points towards a lifelong learning capability. By retrieving and bootstrapping successful strategies, the system becomes more efficient and robust over time, a contrast to the static nature of policies that require extensive fine-tuning or retraining to adapt Kim et al. (2025b); Lee et al. (2025).

### A.3.2 LIMITATIONS AND FUTURE WORK

Despite its promising results, CoRAL has several limitations that define clear directions for future research.

**Fidelity of the Internal World Model:** The entire framework is predicated on the quality of the simulated planning world constructed by the vision module. This dependency is a significant limitation; the performance of the MPPI planner is directly correlated with how accurately this internal model reflects real-world physics. We can think of this simulated planning environment as the robot's "mind," where it mentally rehearses actions before execution. The better this mental model, the more seamlessly the robot can translate its plans into successful evaluation-world actions. Currently, the system is vulnerable to inaccuracies in object pose estimation, and VLM "hallucinations" or inaccuracies—misjudging an object's material could lead to a grossly incorrect estimate for mass or friction. While our feedback loop is designed to correct for such errors, a sufficiently poor initialization could prevent the planner from converging. Future work should focus on creating higher-fidelity internal world model, potentially by learning residual dynamics models online to capture unmodeled effects (e.g., non-rigid dynamics, complex friction) and better bridge the sim-to-real gap.

**Computational Latency:** Vision perception, LLM reasoning, and MPPI planning are the sources of computational latency. In our implementation, the LLM and VLM component uses GPT-4o via the OpenAI API, and a single query takes on average around 1.5 seconds to return a response as an LLM and 2.3 seconds to responds as a VLM. Importantly, the LLM call is not issued at every control step; it is only triggered when the outer loop (e.g., strategy or cost-function/parameter update) is invoked, every 15 control steps. Before execution starts, we make a single VLM call for physical parameter estimation (2.3 s) and perform the initial object registration with FoundationPose (2.1 s), leading to a one-time startup latency of about 4.4 s. During execution, FoundationPose tracking incurs 36 ms per pose update, and MPPI (200 trajectories, horizon 50) takes, on average, 0.4 s per planning iteration, which is the main bottleneck. As a result, a typical inner-loop update without an LLM call has an image-to-command latency of roughly 0.44 s, while an outer-loop update that involves online LLM adaptation or failure recovery adds about 3 s (two GPT-4o calls: one for cost, one for parameters). If a similar episode is found in the memory unit, the initial LLM call is skipped and the rollout starts from a warm cost function, which reduces both the number of LLM queries and the overall latency.

Another important bottleneck arises from the use of simulation environments for planning. MPPI requires rolling out many trajectories in parallel, and while this is embarrassingly parallel in principle, its efficiency is constrained by the underlying physics engine. In our work, we implement MPPI with parallelized environments on the CPU, since MuJoCo currently does not support GPU-based simulation and we aim to remain comparable to VLA baselines that are commonly benchmarked on LIBERO, which is built on top of MuJoCo. In a real-world deployment, or with a GPU-accelerated physics backend, the trajectory rollouts could be significantly faster. More generally, the performance of the MPPI planner can be improved by combining such systems-level optimizations with algorithmic advances, such as more efficient sampling strategies and horizon adaptation Firoozi et al. (2025); Ma et al. (2024); Pezzato et al. (2025), to bring the overall latency closer to what is required for highly dynamic, real-time robotic tasks.

**Reliance on Generalist LLMs for Strategy Formulation:** A core component of our system is the LLM's ability to generate a viable cost function for the MPPI planner. We currently use a generalist, off-the-shelf LLM (GPT-4o), which performs remarkably well but is not specialized for robotics or physics-based planning. The quality and coherence of the generated cost function are not guaranteed for entirely novel or abstract tasks that fall outside the LLM's vast but general pre-training data. A promising direction for future work is to fine-tune or develop LLMs specifically for the task of generating optimization objectives for robotic control, potentially leading to more robust and efficient strategy formulation.

A.4 CONTACT STRATEGY ABLATION

Figure 5 illustrates the results of the LLM-guided contact strategy ablation, detailed in the main text's robustness analysis. To purely isolate the impact of the strategic guidance on the planner, this specific experiment was conducted using known ground truth object poses and sizes, thereby eliminating any potential confounds from the perception module. The figure visualizes the stark difference between the direct, efficient trajectory generated with the LLM's guidance (green) and the erratic, inefficient path taken by the unguided planner (red).

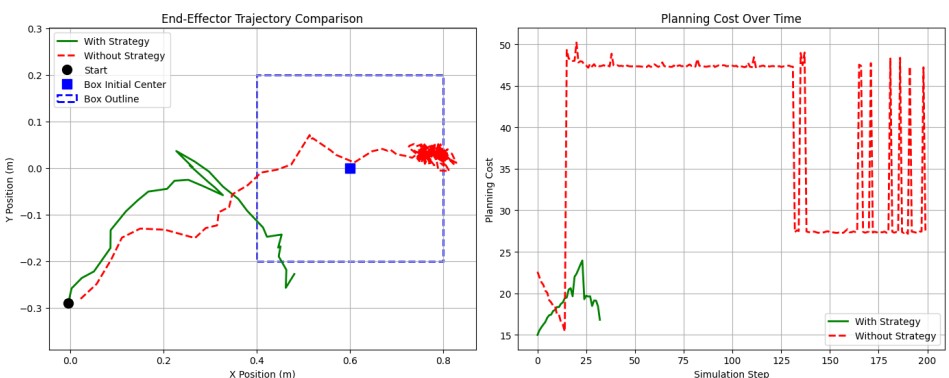

Figure 5: Ablation of the LLM-guided contact strategy on the "Flip with Wall" task. **(Left)** The trajectory with the LLM's strategy (green) is direct and efficient, while the unguided trajectory (red) is erratic. **(Right)** The planning cost for the guided agent is significantly lower and more stable, indicating an easier optimization problem.

A.5 INTERNAL WORLD MODEL FOR PLANNING AND ADAPTATION

The core of CoRAL's reasoning capability lies in its use of an explicit internal world model. This model is a simulated, physics-aware representation of the real world, constructed and parameterized by the vision module's output, $\theta$. This internal model is not merely a passive environment for the planner; it serves two critical functions in our architecture:

1. **A Sandbox for Mental Rehearsal:** The MPPI controller leverages this world model to perform a number of parallel "mental rehearsals" (rollouts) of potential action sequences. This allows the agent to anticipate the physical consequences of its actions—such as contact forces and object displacement—before executing them in the evaluation world, enabling proactive and intelligent decision-making.

2. **An Adaptable Belief State:** This world model represents the agent's current "belief" about the physical properties of its environment. Crucially, this belief is not static. The outer feedback loop, driven by the 'LLM (Online Adaptation)' module, directly refines this model by updating its parameters ($\theta$) based on discrepancies between predicted and observed outcomes. This turns the world model into a dynamic, adaptable belief state that is continuously improved through physical interaction, connecting our work to principles of online system identification and model-based reinforcement learning.

Below is the direct natural language output from the 'LLM (Online Adaptation)' module after being triggered by persistent failures in the "Flip with Wall" task. The LLM was provided with the episode history ($E_t$) and prompted to diagnose the failure and propose a refined cost function.

A.6 EXAMPLE OF AN LLM-GENERATED COST FUNCTION

Figure 1 shows a direct example of a Python cost function generated by the 'LLM (Task Formulation)' module for the multi-stage "Push and Pick Cutting Board" task (T1). This code serves as the

```
--- Outer Loop Triggered:  Analyzing failure and refining strategy...
---

LLM Explanation:  The previous cost function was failing because it did not sufficiently priori-
tize the critical aspects of the task, such as ensuring the box is flipped onto its side and maintaining
contact with the wall. The rewards and penalties were not balanced effectively to guide the robot towards
achieving the task goal. Specifically, the orientation error was not heavily penalized, and the rewards for
contact and elevation were not strong enough to drive the desired behavior.

In the new version, the following improvements have been made:
```

1. **Increased Penalty for Orientation Error**: The weight on the angle error has been increased
   to emphasize the importance of achieving the correct orientation.

2. **Stronger Rewards for Contact and Elevation**: The rewards for maintaining contact with the
   wall and elevating the box have been increased, encouraging the robot to push the box against
   the wall and attempt a flip.

3. **Adjusted Proximity Penalties**: The penalties for the end-effector's distance from the box and
   the box's distance from the wall have been adjusted to better guide the robot's actions.

4. **Relaxed Orientation Threshold**: The threshold for the orientation reward has been slightly
   relaxed to allow for some tolerance in achieving the target orientation, which can help in stabi-
   lizing the box on its side.

```
These changes aim to create a more balanced and effective cost
function that better aligns with the task objectives.  Simulated
environment's cost function has been updated.
```

Figure 6: Direct output from the LLM during an online adaptation cycle. The model provides a
clear, human-readable diagnosis of the previous plan's failure and details the specific changes made
to the cost function for the next attempt.

running cost, $q(x_t, u_t)$, within the MPPI objective (Eq. 4). The function demonstrates the LLM's
ability to translate a complex, sequential goal—"push the board until the handle is over the edge,
then pick it"—into a structured, computable objective that the planner can optimize.

The function skillfully balances multiple, often competing, objectives by implementing a dynamic,
state-aware logic:

- **Dynamic Goal Blending:** It calculates a 'grasp_score' based on the board's stability and
  how far its handle overhangs the table edge. This score is used in a sigmoid function to
  create a "soft switch" that smoothly transitions the robot's objective from pushing the board
  to picking the handle.

- **Phase-Specific Rewards:** The cost function provides rewards tailored to the current phase.
  In the push phase, it rewards progress in moving the board to the edge ('Push progress
  reward'). In the pick phase, it rewards correct gripper alignment and lifting the board
  ('Pick-phase alignment', 'Reward lift after grasp').

- **Stability Constraints:** Throughout the push phase, it heavily penalizes any lateral drift or
  rotation of the board, ensuring a stable push.

- **Contact Management:** It explicitly manages contact, penalizing the gripper for losing
  contact with the board during the push phase.

This example highlights how CoRAL's LLM grounds abstract, sequential instructions into a sophis-
ticated, mathematical objective that enables the execution of complex, long-horizon tasks.

```
1 def state_cost(self):
2     # --- Helper lambdas ---
3     def sigmoid(x): return 1.0 / (1.0 + np.exp(-x))
4     def clip01(v):  return max(0.0, min(1.0, float(v)))
5
6     # --- EE pose ---
7     ee_state = p.getLinkState(self.panda, self.grasp_link,
8                                 computeForwardKinematics=True,
```

```
 9                              physicsClientId=self.cid)
10    ee_pos = np.array(ee_state[0])
11    cost = 0.0
12
13    # --- Board pose/vel ---
14    board_pos, board_ori = p.getBasePositionAndOrientation(self.board,
      physicsClientId=self.cid)
15    board_pos = np.array(board_pos)
16    linvel, angvel = p.getBaseVelocity(self.board, physicsClientId=self.
      cid)
17
18    # --- World/task params with safe defaults ---
19    push_axis = np.array(getattr(self, ``push_axis", [1.0, 0.0, 0.0]),
      dtype=float)
20    push_axis /= (np.linalg.norm(push_axis) + 1e-9)
21    table_edge_x = getattr(self, ``table_edge_x", self.board_init_pos[0]
      + 0.25)
22    desired_overhang = getattr(self, ``desired_overhang", 0.06)
23
24    # --- Contact strategy points ---
25    pick_point  = np.array(getattr(self, ``best_contact_point", board_pos
      ), dtype=float)
26    push_point  = np.array(getattr(self, ``push_contact_point",
27                               board_pos + 0.5 * desired_overhang *
       push_axis), dtype=float)
28
29    # --- Overhang & readiness for grasping ---
30    overhang_m   = np.dot(pick_point - np.array([table_edge_x, board_pos
      [1], board_pos[2]]), push_axis)
31    overhang_nrm = clip01(overhang_m / max(1e-6, desired_overhang))
32    v = np.linalg.norm(linvel) + 0.5 * np.linalg.norm(angvel)
33    stability = clip01(np.exp(-3.0 * v))
34    grasp_score = 0.7 * overhang_nrm + 0.3 * stability
35    r = sigmoid(12.0 * (grasp_score - 0.55))  # soft switch: r -> 1 as
      grasp becomes viable
36
37    # --- Blended target following (Push vs. Pick) ---
38    w_follow_push = 12.0 * (1.0 - r)
39    w_follow_pick = 20.0 * r
40    cost += w_follow_push * np.linalg.norm(ee_pos - push_point)
41    cost += w_follow_pick * np.linalg.norm(ee_pos - pick_point)
42
43    # --- Push progress reward ---
44    cost -= 18.0 * (1.0 - r) * overhang_nrm
45
46    # --- Contact management (for push phase) ---
47    in_contact = bool(getattr(self, ``in_contact", True))
48    cost += (22.0 * (1.0 - r)) * (0.0 if in_contact else 1.0)
49
50    # --- Pick-phase alignment ---
51    ee_quat = ee_state[1]
52    if hasattr(self, ``handle_desired_ori") and self.handle_desired_ori
      is not None:
53        q_diff = p.getDifferenceQuaternion(ee_quat, self.
      handle_desired_ori)
54        w = max(-1.0, min(1.0, float(q_diff[3])))
55        angle = 2.0 * math.acos(w)
56        cost += (28.0 * r) * angle
57
58    # --- Lateral drift & rotation penalties (for push phase) ---
59    dp = board_pos - np.array(self.board_init_pos)
60    axial  = np.dot(dp, push_axis) * push_axis
61    lateral = dp - axial
62    cost += 10.0 * (1.0 - r) * np.linalg.norm(lateral)
63    q_diff_b = p.getDifferenceQuaternion(board_ori, self.board_init_ori)
```

```
64    wb = max(-1.0, min(1.0, float(q_diff_b[3])))
65    ang_b = 2.0 * math.acos(wb)
66    cost += 8.0 * (1.0 - r) * ang_b
67
68    # --- Reward lift after grasp ---
69    lift_h = board_pos[2] - self.board_init_pos[2]
70    cost -= (32.0 * r) * max(0.0, float(lift_h))
71
72    # --- General penalties ---
73    if hasattr(self, ``last_action"):
74        cost += 0.001 * np.linalg.norm(self.last_action) ** 2
75    cost += 0.01 * getattr(self, ``current_step", 0)
76
77    return cost
```

Listing 1: The complete, LLM-generated running cost function $q(x, u)$ for the "Push and Pick Cutting Board" task (T1). It demonstrates a sophisticated, multi-stage logic with a soft switch to blend objectives between the pushing and picking phases.

### A.7 VLM PROMPTING FOR PHYSICAL PARAMETER ESTIMATION

This section details the query sent to the VLM (GPT-4o) to infer the physical properties of objects in the scene, which are used to parameterize the internal world model, $\theta$. Unlike methods that use VLMs for identification, our approach leverages the VLM's physical commonsense reasoning. The VLM's task is not to identify objects, but to estimate their unobservable physical attributes based on their known identity and estimated geometric state.

The prompt, shown in Figure 2, provides the model with all available context: the full scene image (passed implicitly to the multimodal model), the natural language task description $T$, and a JSON object for each relevant item. This JSON object is populated with the object's known semantic 'label' (from its 3D model) and its current 6-DoF 'pose', as estimated by FoundationPose.

The VLM's sole task is **Estimation**: It must use its vast, pre-trained knowledge about the real world to estimate the physical properties of the object, such as its 'mass_kg' and 'friction_coeff', based on its visual appearance (e.g., material, size) and the provided context. By constraining the output to be only the completed JSON object, we ensure the response is directly parsable by our system.

```
1   # Prompt sent to GPT-4o to act as the VLM
2   You are a robotics expert with a deep understanding of physics.
3   Your task is to estimate the physical properties of an object for a
        simulation, based on its appearance in an image.
4
5   Task Description: ``Push the cutting board until the handle is off the
        table, then pick it up."
6
7   I have an object in the scene. I know what it is and I have an estimate
        of its current pose. Please provide your best estimate for its mass (
        in kg) and friction coefficient based on the image.
8
9   Object Data:
10  {
11    ``cutting_board": {
12      ``pose_estimated": [0.5, 0.0, 0.0, 0.0, 0.0, 0.0, 1.0],
13      ``mass_kg": "?",
14      ``friction_coeff": "?"
15    }
16  }
17
18  Respond ONLY with the completed JSON object, filling in the unknown
        values.
19
20  # --------------------------------------------------
21  # Example VLM JSON Response for the task above:
22  {
```

```
23    ``cutting_board": {
24      ``pose_estimated": [0.5, 0.0, 0.0, 0.0, 0.0, 0.0, 1.0],
25      ``mass_kg": 0.4,
26      ``friction_coeff": 0.4
27    }
28  }
29
```

Listing 2: The structured prompt sent to the multimodal model (GPT-4o) to act as our VLM. The model is provided with the object's known label and estimated pose, and is tasked only with filling in the unknown physical parameters (?).

## A.8  LLM PROMPTING FOR COST FUNCTION GENERATION

```
1  You are an expert in optimal control, physics-based planning, and contact
     -rich manipulation.
2  Your job is to generate an initial mppi cost function that adapts to the
     task description
3  and world parameters.
4
5  IMPORTANT: The structure must be GENERAL and TASK-ADAPTIVE.
6  Do NOT hard-code logic for a specific task. Instead, infer the
     requirements from the task description.
7
8  You must output ONLY a Python-like function:
9
10     def state_cost(self):
11
12  STRUCTURAL REQUIREMENTS:
13  1. The cost must be composed of weighted terms for:
14     - distance-to-goal or subgoal
15     - contact or interaction constraints (if relevant)
16     - orientation/alignment terms (if relevant)
17     - force or stability terms (if relevant)
18     - control effort
19     - time/step penalty
20
21  2. If the task involves MULTIPLE PHASES (e.g., push then pick, flip then
      place),
22     you MUST:
23     - infer subgoals,
24     - compute a phase progress score,
25     - optionally blend objectives using a soft switch (sigmoid-based).
26
27  3. If the task involves CONTACT-RICH behavior (e.g., pushing, flipping,
      using a wall),
28     include:
29     - contact incentives or penalties,
30     - drift or slippage penalties,
31     - force-dependent shaping terms (if sensed).
32
33  4. If the task is SIMPLE (e.g., pick-and-place),
34     use a single-stage goal-driven cost with alignment and distance terms.
35
36  5. ALL TERMS must be conditional on task semantics.
37     Only include what is relevant for the input task.
38
39  6. ALL WEIGHTS must be numeric (choose reasonable magnitudes).
40
41  7. The function MUST be fully executable Python-like pseudocode using:
42     - norms
43     - dot products
44     - quaternions (if needed)
45     - sigmoid for soft transitions
```

```
46      - optional heuristics (e.g., stability)
47
48  FORMAT (MANDATORY):
49  Return ONLY the code block:
50
51      def state_cost(self):
52          ...
53          return cost
54
55  --------------------------------------
56  TASK DESCRIPTION:
57  "{TASK_DESCRIPTION}"
58
59  POSE_STATE:
60  {TRACKED_POSES_JSON}
61
62  PHYSICAL_PARAMS:
63  {ESTIMATED_PARAMS_JSON}
64
65  Return ONLY the Python code block.
```

Listing 3: General task-adaptive prompt used by the LLM to generate initial MPPI cost functions.

### A.9   LLM PROMPTING FOR ONLINE ADAPTATION

The 'LLM (Online Adaptation)' module is triggered when the system detects persistent failures. Unlike the initial task formulation, the adaptation prompts are designed to be diagnostic, providing the LLM with a history of recent failed interactions to inform its corrections. The module employs two distinct prompting strategies depending on the type of refinement needed: strategy refinement (correcting the plan) and world model correction (correcting physical parameters).

#### A.9.1   STRATEGY REFINEMENT PROMPT

When the logic of the plan itself is suspected to be flawed, the system asks the LLM to act as a robotics programmer and rewrite the core 'state_cost' function. As shown in Figure 4, the prompt provides the LLM with the task, the environment's attributes, the recent execution history (e.g., last 5 steps of object positions and resulting costs), and critically, the **current, failing source code** of the cost function. It is then instructed to return a corrected code block and a natural language explanation of its changes. This process is the source of the explainable failure recovery analysis presented in the main text.

```
1  # Python function that builds the prompt for cost function refinement.
2  def ask_state_cost_fn(self, task: str, history: list, env: BoxPushEnv):
3      # Grab the current, failing source code of the cost function.
4      try:
5          current_src = inspect.getsource(env.state_cost.__func__)
6      except (OSError, IOError):
7          current_src = env.state_cost_src
8
9      # --- The prompt sent to the LLM ---
10     prompt = (
11         f"Task: {task}\n"
12         "You have full freedom to compute any cost that helps '{task}'.\
    n"
13         "History of last 5 steps (object_pos, resulting_cost):\n"
14         f"<RECENT_EXECUTION_HISTORY>\n\n"
15         "Please output TWO things, separated by '---':\n"
16         "1) Python code for a method 'def state_cost(self): ...' (
    indented block only)\n"
17         "2) A brief explanation why the previous cost was failing and
    how the new one addresses it.\n\n"
18         f"Here is the CURRENT implementation that is failing:\n"
19         f"{current_src}"
```

```
20          )
21
22      # Query the LLM and parse the response (code_str, explanation)
23      resp = openai.chat.completions.create(...)
24      content = resp.choices[0].message.content
25      code_str, explanation = content.split("---",1)
26      return code_str.strip(), explanation.strip()
27
```

Listing 4: The Python function and prompt structure used for Strategy Refinement. The LLM is given the failing code and recent history to rewrite the cost function.

### A.9.2 WORLD MODEL CORRECTION PROMPT

If the strategy is believed to be correct but the physical outcomes do not match the simulation (e.g., the robot pushes but the object barely moves), the system asks the LLM to act as a physicist and refine the object parameters. The prompt, detailed in Figure 5, provides the LLM with the agent's current belief about the physical parameters (mass, friction) and the recent execution history. The LLM's task is to analyze the discrepancy between actions and outcomes in the history and propose corrected physical parameters, returning them in a machine-parsable JSON format.

```
1  # Python function that builds the prompt for physical parameter
       refinement.
2  def ask_params_refinement(self, history: list, object_params: list):
3
4      # --- The prompt sent to the LLM ---
5      prompt = (
6          ``We have the following object parameters:\n"
7          f"{json.dumps(object_params, indent=2)}\n\n"
8          ``Recent execution history (last 5 steps):\n"
9          f"<RECENT_EXECUTION_HISTORY>\n\n"
10         ``Based on this, propose refined values for mass and
       friction_coef."
11         ``Return ONLY a JSON array of objects with keys "
12         "`label', `mass', and `friction'.\n"
13         ``Example output:\n"
14         "[\n"
15         "   {\``label\":\``cutting_board\",\``mass\":0.45,\``friction
       \":0.35}\n"
16         "]"
17     )
18
19     # Query the LLM and parse the JSON response
20     resp = openai.chat.completions.create(...)
21     text = resp.choices[0].message.content.strip()
22     try:
23         return json.loads(text)
24     except json.JSONDecodeError:
25         # Fallback to extract JSON if LLM adds extra text
26         ...
27
```

Listing 5: The Python function and prompt structure for World Model Correction. The LLM analyzes the recent history to refine its belief about the object's physical properties.

### A.10 LLM-DRIVEN CONTACT STRATEGY GENERATION

A key challenge in contact-rich manipulation is determining precisely *where* to make contact with an object. Uniformly sampling an object's entire surface is computationally inefficient and unlikely to yield strategically useful points. To overcome this, CoRAL leverages the LLM's commonsense physical reasoning to intelligently narrow the search space. This is achieved through a two-stage process, detailed in Figure 6, which translates a high-level task into a concrete set of candidate contact points ($C_0$).

**Stage 1: Strategic Region Proposal.** First, the 'LLM (Task Formulation)' module queries a foundation model (GPT-4o) with a structured prompt that includes the task description and the VLM-estimated object parameters. The prompt, shown in Figure 7, explicitly instructs the LLM to act as a robotics expert and identify 1-3 small, promising surface regions for contact on each relevant object. The LLM is constrained to return this information in a structured JSON format, specifying each region's 3D center, surface normal, radius (extent), and the desired number of samples. For the "Push and Pick Cutting Board" task, the LLM correctly identifies that pushing should occur on the board's main surface, while grasping should target the handle.

**Stage 2: Geometric Candidate Sampling.** Second, the structured JSON response from the LLM is passed to a geometric sampling function ('sample-points-in-region'). This function translates the LLM's abstract region definitions into a dense set of 3D point coordinates. For each region, it defines a 2D disk in 3D space oriented by the provided center and normal vectors. It then samples the requested number of points within this disk, generating the final set of candidate contact points, $C_0$, which are then used to bias the MPPI planner's exploration as described in the main text.

```python
1  # Stage 1: Query the LLM to propose strategic contact regions.
2  def ask_region_strategy(object_params, task_desc):
3      # The prompt is shown in Figure~\ref{fig:contact_prompt}
4      prompt = f"..."
5      resp = client.chat.completions.create(...)
6      return resp.choices[0].message.content
7
8  # Stage 2: Sample concrete 3D points from an LLM-defined region.
9  def sample_points_in_region(region):
10     c = np.array(region[``center"], dtype=float)
11     n = np.array(region[``normal"], dtype=float)
12     r = float(region[``extent"])
13     k = int(region[``num_samples"])
14
15     # Create two orthonormal tangent vectors to define the plane of the
       disk.
16     if abs(n[2]) < 0.9:
17         axis = np.array([0.0, 0.0, 1.0])
18     else:
19         axis = np.array([0.0, 1.0, 0.0])
20     t1 = np.cross(n, axis)
21     t1 /= np.linalg.norm(t1)
22     t2 = np.cross(n, t1)
23     t2 /= np.linalg.norm(t2)
24
25     # Sample k points within the 2D disk defined by the tangents.
26     pts = []
27     for _ in range(k):
28         rho   = np.sqrt(np.random.rand()) * r  # Uniform sampling in a
       disk
29         theta = np.random.rand() * 2 * np.pi
30         offset = rho * np.cos(theta) * t1 + rho * np.sin(theta) * t2
31         pts.append((c + offset).tolist())
32     return pts
33
```

Listing 6: The two-stage Python implementation for generating the contact strategy $C_0$. The 'ask-region-strategy' function queries the LLM for high-level guidance, and 'sample-points-in-region' translates that guidance into concrete 3D coordinates.

```
1  # The prompt sent to the LLM (Task Formulation) module:
2  Task: Push the cutting board until the handle is off the table, then pick
       it up.
3
4  You have objects with parameters:
5  {
6
7    ``board": [{TRACKED_POSES_JSON} , {ESTIMATED_PARAMS_JSON}],
```

```
 8    ''table": [{TRACKED_POSES_JSON} , {ESTIMATED_PARAMS_JSON}]
 9  }
10
11  Instead of uniform sampling, identify for each object 1-3 small surface
       regions
12  where contact is most promising.
13  For each region, return:
14    - center: [x,y,z]
15    - normal: unit surface normal [nx,ny,nz]
16    - extent: radius in meters around center
17    - num_samples: how many points to sample there
18
19  Respond ONLY a JSON mapping each label to its ''regions" array.
20
21  # Example LLM JSON Response:
22  {
23    ''object": {
24      ''regions": [
25        {
26          ''center": [0.5, 0.0, 0.01],
27          ''normal": [0, 0, 1],
28          ''extent": 0.1,
29          ''num_samples": 30
30        }
31      ]
32    }
33
```

Listing 7: The structured prompt and an example JSON response for the "Push and Pick Cutting Board" task. The prompt constrains the LLM to provide a structured, machine-readable output.