# OpenReview forum: "Contact-VLA: Zero-Shot Planning and Control for Contact-Rich Manipulation"
_ICLR.cc/2026/Conference — Submitted to ICLR 2026_

### Official Review · Reviewer_Nicm · 2025-10-30

**Soundness:** 2
**Presentation:** 3
**Contribution:** 2
**Rating:** 2
**Confidence:** 4

**Summary:**

The paper presents Contact-VLA, a novel framework for zero-shot planning and control in contact-rich manipulation tasks. It integrates Vision-Language-Action (VLA) systems with a reactive motion controller, leveraging Large Language Models (LLMs) for high-level planning, strategy formulation, and online adaptation. The core idea is to combine the broad understanding of VLAs/LLMs with the precision and robustness of reactive control for dynamic, contact-heavy robotic interactions. The authors claim that Contact-VLA significantly outperforms prior state-of-the-art methods on various manipulation tasks, validated through experimental comparisons, ablation studies, and robustness analysis. The framework's ability to perform zero-shot tasks and recover from failures is a key contribution.

**Strengths:**

1.  The paper addresses a significant challenge in robotics by proposing a unified framework that combines the semantic understanding of LLMs/VLAs with the fine-grained control needed for contact-rich manipulation. The framework demonstrates promising zero-shot planning and control, reducing the need for extensive task-specific data and potentially enabling rapid deployment in new scenarios.
2. The inclusion of mechanisms for online adaptation and LLM-driven failure analysis and refinement (e.g., modifying cost functions, adjusting penalties) significantly enhances the system's robustness to uncertainties and unexpected events during execution.
3. The comparative results against state-of-the-art baselines (e.g., OpenVLA-OFT on the LIBERO benchmark) show a clear performance advantage for Contact-VLA in terms of success rate and completion time. The ablation studies provide insights into the contributions of key components, such as the LLM-guided contact strategy, which helps validate design choices.

**Weaknesses:**

1. The paper acknowledges computational latency as a limitation. While a common issue, more discussion on specific strategies to mitigate this for real-time robotic applications, beyond general statements, would be beneficial.
2. The reliance on LLMs for strategy formulation, while enabling zero-shot capabilities, raises questions about the consistency, optimality, and safety guarantees of these generated strategies across a truly diverse and potentially adversarial set of contact scenarios.
3. The effectiveness of the reactive control and adaptation hinges on an accurate internal world model and physical parameter estimation. More details on how these are learned, maintained, and their potential failure modes would strengthen the paper.

**Questions:**

1. Could the authors provide more specifics on the prompt engineering or fine-tuning techniques used to guide the LLM for generating cost functions, refining strategies, and performing failure analysis? How sensitive is the system to variations in these prompts?
2. What is the end-to-end latency of the Contact-VLA system, from visual input to motor commands, particularly when the LLM is involved in an online adaptation or failure recovery loop?
3. Are there any formal guarantees or theoretical bounds on the stability and safety of the reactive controller when its parameters and objectives are dynamically modulated by an LLM?
4. How does Contact-VLA handle situations where the LLM's generated strategy is suboptimal, unsafe, or leads to unexpected physical interactions not covered by its training data?
5. Can the authors elaborate on the Internal World Model and Physical Parameter Estimation? How are these models learned, updated, and how robust are they to novel object properties or environmental changes?
6. What is the current compute cost (e.g., GPU hours, energy consumption) for training and deploying Contact-VLA, especially considering the LLM components?
7. Beyond the tasks presented, what are the current limitations of Contact-VLA in terms of object complexity, number of contact points, or task duration?
8. Can the authors provide specific examples of how Contact-VLA offers improved explainability compared to other end-to-end learning systems for manipulation?

---

> ### Author Response · Authors · 2025-11-21
> **Addressing Latency, LLM Guarantees, and World Model Robustness**
>
> **Weakness-1**
>
> We have extended the Computational Latency part in Section A.3.2 (Limitations and Future Work) to discuss concrete mitigation strategies(lines 778-802). In particular, we now detail: (i) the current latency of our GPT-4o-based LLM calls and how this could be reduced via local deployment or smaller, robotics specific models, and (ii) how MPPI latency can be mitigated through parallelization, GPU-accelerated physics backends, and more efficient sampling strategies for trajectory rollouts.
>
> **Weakness-2**
>
> We agree that relying on an LLM for strategy formulation raises questions about consistency, optimality, and safety, and we do not claim any formal guarantees in this work. Our evaluation focuses on demonstrating that LLM-guided strategies are practically useful and robust across diverse initializations, as supported by our ablation and robustness experiments.
>
> We also note that our VLA baselines likewise do not provide theoretical guarantees or explicit explanations of why particular actions are chosen. In contrast, our pipeline is more transparent: it exposes an explicit cost function that directly links behavior to optimization objectives, and this cost function is generated (and can be explained) by the LLM. This makes it possible to inspect which terms drive a given behavior and to obtain natural-language rationales for the chosen strategy, improving explainability relative to standard end-to-end VLA models.
>
> We view the development of formal guarantees for LLM-based planners as a critical but distinct open research area. Establishing theoretical bounds on consistency and safety for large foundation models is a grand challenge facing the entire field, and addressing it is orthogonal to the primary contribution of our work. Our focus remains on demonstrating that language-guided, physics-based adaptation can empirically solve complex tasks that current data-driven baselines cannot.
>
> **Empirical Assessment of Optimality:** While formal guarantees are challenging, we will be addressing the practical question of optimality by introducing a "Handcrafted Multi-Phased Baseline" (expert human design) in the revision. Benchmarking the LLM against this expert baseline will allow us to quantitatively assess whether the generated strategies diverge significantly from expert-level solutions. Please see Point 7 in our General Response for details.
>
> **Weakness-3**
>
> **Internal World Model:** In our framework, the internal world model captures a small set of task-relevant physical parameters: the masses of the interactable objects and the friction coefficients of relevant surfaces (e.g., table, floor, wall). These parameters are initially estimated by querying a VLM with an image of the scene and the prompt described in Appendix A.9. During execution, they are periodically updated by the LLM based on the interaction history (see Fig. 2).
>
> **Potential Failure Modes:** A key failure mode of this pipeline is that purely visual cues can be misleading. First, visual ambiguity of materials can cause the VLM to infer incorrect typical mass or friction values. Second, lighting, reflections, and texture confusion may distort surface appearance. Third, occlusions and partial views can force the VLM to extrapolate from incomplete evidence. Finally, many real-world objects are heterogeneous, so a single global estimate may not capture the local properties of the contact region.
>
> **Mitigation and Comparison to VLAs:** To mitigate these issues, our framework does not treat initial VLM estimates as fixed. Instead, the LLM periodically updates physical parameters based on observed mismatches between predicted and actual outcomes. As demonstrated in our robustness analysis (Section 4.1.3, Figure 3), this online adaptation loop allows the system to recover from severely biased initial estimates.
>
> Crucially, this capability offers a distinct advantage over standard VLA models. VLAs rely on implicit physical priors encoded in their weights during training, which remain frozen during inference. Consequently, if a test object violates these learned priors (e.g., a surface that looks rough but behaves slippery), VLAs typically suffer catastrophic failure with no mechanism for recourse. In contrast, our framework actively detects such discrepancies and updates its explicit world model on the fly, enabling it to recover from perceptual errors and succeed where static, end-to-end policies would fail.

---

> ### Author Response · Authors · 2025-11-21
> **Prompt Engineering, Sensitivity, and Latency Analysis**
>
> **Question-1**
>
> **Prompt Specifics and Sensitivity:** We have added the full prompt used for the initial cost function generation to Appendix A.10 for completeness. Previously, this prompt was not listed separately because it closely resembles the cost-refinement prompt, and we mistakenly assumed that including only the refinement prompt would suffice to convey the overall process. The VLM prompt for physical parameter estimation (Appendix A.9), the strategy refinement prompt (Appendix A.11.1), the world model correction prompt (Appendix A.11.2), and the contact strategy generation prompt (Appendix A.12) were already included in the original submission.
>
> All of these prompts follow a generic template in which the task description, tracked object poses, and estimated physical parameters are dynamically instantiated. Because the structure of the prompts is fixed and only these instantiated fields change, the system is fairly robust to prompt variations; the main failure modes arise when the pose tracker produces highly inaccurate poses or when the provided task description is severely misaligned with the actual task.
>
> **Comparison to Related Work:** We contrast our approach with concurrent work such as ReKep (Huang et al., 2024). While ReKep also leverages VLMs for zero-shot planning, it relies on a highly engineered and constrained prompting strategy to generate executable Python code. As detailed in their Appendix A.6, the ReKep prompt enforces strict syntactic rules such as prohibiting "if" statements , enforcing specific function signatures for "sub-goal" vs. "path" constraints , and hard-coding logic for specific action verbs (e.g., explicit rules distinguishing "grasping" stages from "non-grasping" stages ). Furthermore, their prompts rely on extensive task-specific examples (e.g., specific logic for pouring tea or folding garments ) to guide the output.
>
> In contrast, our framework does not require the LLM to adhere to such a rigid, action-specific code structure or a complex set of syntactic constraints. Instead, we use a flexible natural language template that focuses on translating semantic goals into optimization terms. This allows our system to be less sensitive to the specific phrasing of "verbs" or pre-defined action primitives, offering a more generalizable approach to cost generation compared to the heavily structured prompt engineering required by methods like ReKep.
>
> **Question-2**
>
> Before execution starts, we make a single VLM call for physical parameter estimation,taking about 2.3 s, and perform the initial object registration with FoundationPose, which takes about 2.1 s, for a one-time startup latency of roughly 4.4 s before the first command.
>
> During execution, FoundationPose runs in tracking mode, with each pose-update taking about 36 ms, and MPPI (200 trajectories, horizon 50) takes on average 0.4 s per planning iteration on CPU-parallel MuJoCo, which is the main computational bottleneck. The LLM (GPT-4o via the OpenAI API) is not called at every control step: it is queried roughly every 15 control steps, i.e., in the outer loop for strategy/cost updates or failure recovery. Each such LLM call adds about 1.5 s of latency.
> Putting this together, a normal inner-loop update without an LLM call has an image-to-command latency on the order of ≈0.44 s (pose tracking + MPPI), while an update that involves online LLM adaptation or failure recovery has a latency of roughly 3 s (pose tracking + MPPI + LLM) as the LLM is queried once for updating the cost function and once for updating the physical parameters. If a similar episode is found in the memory unit, the initial LLM call is skipped, and the rollout starts from a warm cost function, which both reduces the total number of LLM queries.
>
> We have added these latency details to Appendix A.3.2 under Computational Latency in the revised version.
>
> **Optimization Potential:** We emphasize that these reported latencies reflect a research implementation focused on verifying architectural efficacy rather than an optimized production system. Two key factors currently inflate the runtime but are addressable in future iterations:
>
> **LLM Overhead:** We currently use a very large generalist model (GPT-4o) via a remote API, which introduces significant network and inference latency. In a deployed system, this could be replaced by a smaller, specialized model (distilled from the generalist one) running locally to drastically reduce inference time when an LLM call is present.
>
> **MPPI Acceleration:** Our MPPI planner currently runs on CPU. This design choice was necessitated by our benchmarking against VLAs in the LIBERO environment, which builds on standard MuJoCo and does not trivially support the massive parallelization required for GPU-accelerated simulation. However, in a standalone deployment, transitioning to GPU-based MPPI would likely increase planning frequency from $\approx2.5$ Hz to $>25$ Hz, significantly tightening the control loop.

---

> ### Author Response · Authors · 2025-11-21
> **Formal Guarantees, Failure Recovery, and Internal World Model Details**
>
> **Question-3**
>
> Are there any formal guarantees or theoretical bounds on the stability and safety of the reactive controller when its parameters and objectives are dynamically modulated by an LLM?
> We explicitly acknowledge that our current framework does not provide formal guarantees or theoretical bounds on stability. The probabilistic nature of LLMs makes deriving such guarantees a non-trivial challenge that remains an open problem for the broader field of generative language.
> While our work focuses on empirical robustness and the architectural novelty of zero-shot adaptation, we agree that formal verification is a critical direction for real-world deployment. We view this as an exciting and important area for future research, where our adaptive pipeline could potentially be augmented with safety mechanisms such as Control Barrier Functions (CBFs), Reachability Analysis, or Runtime Verification wrappers to filter LLM-generated costs or parameters before execution. We hope our empirical findings provide a motivating foundation for theoretical explorations.
>
> **Question-4**
>
> In our current implementation, we treat the LLM as a black-box prior for strategy generation. Since we use a generalist model (GPT-4o) via an API, we cannot formally characterize its behavior under all adversarial inputs. However, our framework handles suboptimal or unsafe strategies through a reactive refinement loop.
>
> Unlike open-loop systems that blindly execute a fixed plan, CoRAL periodically re-queries the LLM (e.g., every 15 steps) with the recent interaction history. This allows the model to review the robot's actual performance against the intended goals. If the history reveals that the strategy is yielding unsafe or suboptimal results (e.g., object slippage or unexpected collisions), the LLM generates a refined cost function or updated physical parameters in the next cycle to correct the behavior and recover the task.
>
> Furthermore, due to the modular nature of our framework, we can seamlessly integrate any LLM. The reliability of the system is largely a function of the foundation model used; while we currently employ a generalist LLM for zero-shot capabilities, utilizing a model specifically fine-tuned for robotic manipulation, physical understanding, and safety constraints would serve as a much stronger prior.
>
> **Question-5**
>
> Please refer to our response to Weakness 3, where we provide a detailed explanation of how the internal world model is defined, how mass and friction are estimated and updated, and we discuss robustness and potential failure modes under varying object properties and environmental conditions.

---

> ### Author Response · Authors · 2025-11-21
> **Computational Costs, Deployment, and Limitations**
>
> **Question-6**
>
> **Training and Inference:** We do not train any large models in CoRAL. Crucially, compared to end-to-end VLA policies, which typically require training on massive datasets across multiple GPUs for hundreds of hours, our framework incurs zero training cost. Instead, it leverages the generalization of previously trained foundation models, limiting the GPU computational footprint strictly to inference.
>
> **Deployment Costs:** Both the VLM and LLM components are GPT-4o accessed via the OpenAI API, and the pose tracker (FoundationPose) is used with publicly released pretrained weights. In deployment, GPT-4o is queried only in the outer loop (for initial cost generation, strategy refinement, and parameter updates), with each call taking roughly 1.5 seconds on average; it is not called at every control step. FoundationPose runs locally and is relatively lightweight at inference compared to the rest of the pipeline. The dominant computational cost in our setup comes from MPPI, which runs on CPU with parallel MuJoCo simulations, where the cost scales with the number of sampled trajectories and horizon length rather than GPU usage.
>
>
> **Question-7**
>
> **Object Complexity:** The framework relies on constructing a high-fidelity internal simulation (using MuJoCo).  The framework is currently tested on rigid objects. It can potentially struggle with deformable objects or complex dynamics (e.g., fluids, soft materials) where unmodeled effects create a significant gap between the internal planner and reality. Extending the framework to handle such non-rigid dynamics, potentially via learned residual models or particle-based simulators, is a key direction for future work.
>
> **Task Duration & Dynamics:** While the system handles multi-stage tasks via replanning, the computational latency of the sequential pipeline (Vision $\rightarrow$ LLM $\rightarrow$ MPPI) limits its applicability to highly dynamic tasks requiring millisecond-level reaction times.
> In addition to the limitations discussed in the paper, CoRAL currently relies on accurate pose tracking; significant perceptual drift or partial occlusion can degrade the quality of MPPI rollouts and reduce the reliability of cost-function refinements. Additionally, although the system exhibits strong zero-shot generality, the LLM’s reasoning remains sensitive to fidelity of the provided world-state inputs.
>
> Nevertheless, the goal of this work is not to solve full general robotic manipulation, but to demonstrate that VLM/LLM-based task reasoning can be robustly integrated with model-based controllers to produce contact-rich behaviors without demonstrations. The modularity and online adaptability of our pipeline provide a clear path toward addressing these limitations for example via richer perception models, higher-fidelity physics estimation, or structured prompt templates. We believe the hybrid V/LLM + MPC paradigm introduced here establishes a solid foundation for future systems capable of handling more complex objects, contact configurations, and task horizons.

---

### Official Review · Reviewer_EiND · 2025-11-01

**Soundness:** 2
**Presentation:** 1
**Contribution:** 2
**Rating:** 2
**Confidence:** 2

**Summary:**

The method has not been evaluated on a real-world robotic platform, which undermines the claims regarding contact-rich manipulation, as simulators often struggle to accurately model physical contact dynamics.

**Strengths:**

The authors address the problem of explainability in VLA systems, which is a highly relevant topic for the research community. The proposed method demonstrates superior performance compared to baseline approaches on the evaluated tasks. Furthermore, the authors provide the implementation code.

**Weaknesses:**

The paper exceeds the 9-page limit and, in accordance with the ICLR submission guidelines, should therefore be desk rejected. It also contains several formatting issues, most notably in Figure 3. Furthermore, the plan component in *Equation (1)* is not clearly defined.

The method has not been evaluated on a real-world robotic platform, which undermines the claims regarding contact-rich manipulation, as simulators often struggle to accurately model physical contact dynamics.

Beyond the separation of the VLA into distinct vision and language modules and the introduction of a Memory Unit, the novelty of the contribution appears limited.

**Questions:**

See Weaknesses.

---

> ### Author Response · Authors · 2025-11-21
> **Page Limit Compliance, Real-World Applicability, and Novelty Statement**
>
> **Weakness-1**
>
> We apologize for exceeding the 9-page limit in the original submission. This was caused by a formatting issue related to Figure 3, and we were unfortunately not aware of it at the time of submission. In the revised manuscript, this issue has been fixed and the paper now complies with the 10-page limit specified by ICLR for the revised version. To the best of our knowledge, there are no remaining formatting issues beyond the reviewer pointed out.
>
> **Explanation of Equation (1):** We have also revised Equation (1) and clarified the meanings of $J_{\textrm{mem}}$​ and $C_{\textrm{mem}}$​ in the text. Specifically, $J_{\textrm{mem}}$ denotes the final cost function that led to a successful episode, and $C_{\textrm{mem}}$​ denotes the corresponding contact strategy, which likewise resulted in successful completion of the task.
>
> **Weakness-2**
>
> We agree that evaluation on a physical robot is an important next step, and we have acknowledged this in our Limitations section. However, we maintain that our simulation results provide strong evidence for the method’s suitability for contact-rich manipulation. All tasks in our suite (e.g., pushing with constant force, flipping a box, and flipping against a wall) are explicitly designed to stress contact dynamics, requiring strategies far beyond simple kinematic pick-and-place.
>
> Crucially, our architecture offers inherent resilience to the discrepancies found in real-world deployment. The use of generalist Vision-Language Models (VLMs) helps bridge the perceptual gap by generalizing across both photorealistic and simulated imagery for world building. Furthermore, the reactive capabilities of the LLM allow the system to compensate for dynamic mismatches between real-world physics and the MuJoCo simulation. This is supported by our robustness experiments, where the system successfully diagnosed and corrected severe parameter discrepancies (e.g., incorrect mass and friction), demonstrating the exact type of adaptation mechanism required to mitigate imperfect modeling in the physical world.
>
> Finally, the core contribution of our work is the introduction of a novel neuro-symbolic framework that leverages foundation models to construct and adapt a planning world model. This framework is agnostic to whether the evaluation world is a simulator or a physical robot, provided that standard sensory channels and a reasonably accurate dynamics model are available. We therefore view real-world experiments as an important extension of this work, and a natural direction for future research rather than a fundamental obstacle to the contributions demonstrated in our work.
>
> **Weakness-3**
>
> We respectfully disagree that our contribution is limited to architectural modularity. Our key novelty is the formulation of a zero-shot, language-guided control framework that bypasses the data requirements of traditional VLAs. Specifically, we introduce a mechanism where an LLM acts as a "cost designer," translating high-level semantic goals into dense, continuous cost landscapes for MPPI. This enables the execution of complex, contact-rich behaviors (like force regulation and pivoting) that purely end-to-end models struggle to learn without massive datasets. This synergy of foundation models with physics-based planning, augmented by online parameter adaptation, is a novel paradigm for solving long-horizon manipulation tasks.
>
> If our response does not fully change your assessment, could you please provide more detail on why you view the novelty as limited? Specific feedback would help us understand if we are missing key references or if there is a misunderstanding of our method's capabilities.

---

### Official Review · Reviewer_C4Vf · 2025-11-02

**Soundness:** 2
**Presentation:** 2
**Contribution:** 2
**Rating:** 2
**Confidence:** 3

**Summary:**

The paper proposes a method for contact-rich manipulation by leveraging foundation vision models and large language models in a zero-shot manner. Specifically, given a task, the method first uses LLMs to propose contact plans and cost functions, which are then used for planning with MPPI. To model environment dynamics, physical parameters are initially estimated by the LLM and subsequently updated during plan execution. Based on the rollouts, both the plans and physical parameters are iteratively refined until the task is completed. Experiments demonstrate the effectiveness of the method on six manipulation tasks in the LIBERO environment.

**Strengths:**

1. The paper is easy to read.
2. The idea of using LLM to write cost functions for MPPI is interesting.
3. Experiments have shown the effectiveness of each module.

**Weaknesses:**

1. The tasks considered are very simple, only covering short-horizon basic manipulation (pushing / grasping / pivoting) of simple shapes, and it seems the method assumes access to a list of motion primitives, instead of directly doing joint position / torque control. This further simplifies the problem.
2. The authors assume that all the objects considered have known geometry. This is a strong assumption and greatly limits the generalizability of the method. How can the method generalize to unknown objects or non-rigid objects?
3. The applicability of the physical parameter estimation part is questionable. The authors assume that only 2 values — mass and friction — of a single object need to be estimated. This is another strong assumption: the difficulty of physical parameter estimation grows combinatorially with number of objects or degrees of freedom. Even for the simple case discussed in the paper, there seem to be no physical parameter variations for each task. This does not systematically show the effectiveness of the module. On the other hand, in the real world, online adaptation might be infeasible for hard-to-reset or safety-critical scenarios.

Overall, the method relies on several strong assumptions about the task and environment, but the tasks demonstrated are still relatively simple. Thus I believe it is not meeting the bar of the venue.

**Questions:**

1. In Section 3.2, where do the successful experience episodes come from?
2. What is the prompt for generating the initial cost function and contact strategy?
3. How are the baseline models finetuned on the LIBERO dataset before testing?
4. What is the action space MPPI works on? From the provided code implementation, it seems the method assumes a list of motion primitives to be given for each task. How are they constructed?

---

> ### Author Response · Authors · 2025-11-21
> **Task Difficulty and Absence of Motion Primitives**
>
> **Weakness-1**
>
> **Task Simplicity/Complexity:**
> We agree that standard pick-and-place problems are relatively simple, and we explicitly use T2–T3 as baseline tasks.
>
> However, to empirically contextualize the difficulty of the remaining tasks (T1, T4, T5, T6), we emphasize the performance of the VLA baselines evaluated in our study. Despite being trained on the LIBERO-Goal dataset these models achieved near-zero success rates on our core tasks (as shown in Table 1). This inability of state-of-the-art, data-driven models to solve these tasks demonstrates that they present significant challenges in terms of contact dynamics and precision that are non-trivial for current systems.
> The specific complexities of these tasks are detailed below:
>
> **T1 – Push and Pick Cutting Board:** This task requires the robot to (i) stably push a thin cutting board across the table until a sufficient portion of the board overhangs the edge, and then (ii) transition into a precise grasp and lift. The strategy must reason about where to make contact, how far to push to expose a graspable part, and how friction affects the final pose. Small errors in the first phase accumulate and lead to failure in the second phase, which is reflected in the complete failure of the “w/o Refinement” ablation (0/10 success).
>
> **T4 – Push with Constant Force:** Here the goal is not merely to move the object but to maintain a target contact force while pushing along a trajectory. This requires continuous regulation of interaction forces over a rollout, combining end-effector motion with force feedback control.
>
> **T5 – Flip Box:** In these tasks the robot must generate a controlled rotation of the box around its edge, using pushes that create sufficient torque without overshooting or losing contact. Successful execution is highly sensitive to where on the box the contact is made and how much force is applied, because different friction coefficients and object masses lead to qualitatively different motion. The policy therefore has to reason about contact region and force profile based on the estimated friction coefficient and the mass. Our ablations show that without online refinement, success drops sharply, indicating that the task exposes errors in the world model and requires adaptation rather than simple one-step pushing.
>
> **T6 – Flip with Wall:** This is explicitly designed as a multi-contact, long-horizon task. The robot must first push the box into stable contact with the wall, then exploit the wall as a fixture to pivot the box into the target orientation. This requires coordinating contacts between the gripper, the box, and the wall.
>
> Overall, while two tasks are intentionally simple pick-and-place benchmarks, the core tasks we introduce (T1, T4, T5, T6) require multi-phase, contact-rich, and non-myopic strategies. They expose long-horizon dependencies, sensitivity to physical parameters, and the need for online refinement and experience reuse, which is precisely what our framework is designed to address.
>
> **Empirical Investigation of Task Complexity:** To empirically contextualize the complexity of these tasks, we will be introducing a **"Handcrafted Single-Phase Baseline"** in the revision. We aim to use this baseline to investigate whether tasks like T1 and T6 strictly require the multi-phase reasoning provided by CoRAL, as opposed to simple monolithic objectives. Additionally, we are adding rollout videos of the baseline VLAs to visually illustrate their specific failure modes during contact-rich interactions. Please see Point 7 in our General Response.
>
> **Motion Primitives:** Our implementation **does not** assume any predefined motion primitives, and we do indeed have a controller. Control is determined directly by the model-predictive controller (MPPI in our case), which operates in a continuous end-effector action space rather than over a discrete set of skills. Concretely, MPPI optimizes over a 7D action space consisting of 6D continuous end-effector pose deltas (3D translation and 3D rotation) and a binary gripper command (open/close). MPPI returns action deltas, which are integrated into the current end-effector pose and realized by the low-level controller in robosuite. As a result, force control is handled implicitly through the commanded end-effector positions and the resulting contact interactions in MuJoCo, rather than through explicit force or torque control.

---

> ### Author Response · Authors · 2025-11-21
> **Generalization to Unknown Objects and Physical Parameter Estimation**
>
> **Weakness-2**
>
> **Generalization to Unknown Objects:** We agree that assuming known object geometry is a limitation of the current experimental setup; however, this assumption is not essential to our framework. In fact, it can be relaxed by combining our method with existing object reconstruction and model-free pose estimation techniques. For unknown rigid objects, one can first build an approximate 3D representation from multi-view RGB-D observations (e.g., via NeRF-style reconstruction or related approaches) and then feed this representation to a pose estimator and the simulator.
>
> Our pipeline is already compatible with such modules: the main machinery of our pipeline requires access to object poses. For example, the model-free mode of FoundationPose can derive an internal object representation directly from RGB-D images captured from multiple viewpoints and subsequently use this representation to estimate 6D poses, without a pre-defined geometry model. Integrating this type of model-free perception would allow our method to handle previously unseen rigid objects without changing the core algorithm. We omitted this implementation in our current work, as we consider it as a straightforward extension that is orthogonal to our primary contributions.
>
> **Non-Rigid Objects:** Non-rigid and highly deformable objects are more challenging because they require an appropriate deformable-physics model and corresponding perception and tracking modules. This is orthogonal to the main contribution of our work, which focuses on LLM/VLM-guided contact strategy and cost shaping on top of estimated physical parameters, and we view extension to deformable-object manipulation as an interesting direction for future work rather than a limitation of the proposed framework itself.
>
> **Weakness-3**
>
>
> **Physical parameters:** For the specific contact-rich manipulation tasks we consider (grasping, pushing, pivoting, and flipping rigid rectangular or cylindrical objects on planar surfaces). The dominant physical factors that affect the resulting motion are the mass of the movable objects and the friction coefficients at the relevant contact surfaces (objects, table, floor, and wall). We therefore focused our physical parameter estimation module on these quantities, as we did not identify other physical parameters that would significantly influence the behaviors.
>
> **Parameter variation:** We thank the reviewer for pointing this out. In fact, in our experimental setup the object mass and surface friction coefficients and the box dimensions were already varied across episodes. For each rollout, these parameters were independently sampled from predefined intervals, in addition to the randomized initial object poses. However, we realized that this variation in mass, friction and box dimension was not stated in the original submission. In the revised version, we have made this explicit in the Experiments section under Tasks and Evaluation Metrics (lines 336-337). Specifically, we detail that box density is sampled from $[200, 800] \text{ kg/m}^3$, surface frictions are sampled from $[0.3, 0.8] $ and the box dimensions are sampled from $[0.12, 0.26] \text{m}$.
>
> We agree that in more complex scenes one could consider a higher-dimensional parameter space (e.g., anisotropic friction, restitution, compliance, or multiple interacting objects), and that the difficulty of estimation would increase accordingly. However, we view this as orthogonal to our main contribution, which demonstrates that language-guided physical parameter estimation and adaptation yield immediate benefits even in lower-dimensional spaces. Extending the approach to richer parameter spaces is an interesting direction for future work.
>
> **Regarding specific parameters:** If the reviewer has specific physical properties in mind that are critical for the current set of tasks but are currently unaddressed, we would welcome those suggestions.
>
> **Regarding 'degrees of freedom':** Could you kindly clarify this term in the context of your comment? This clarification would help us better address your concern.

---

> ### Author Response · Authors · 2025-11-21
> **Response to Questions: Memory Unit, Prompts, Baselines, and MPPI Action Space**
>
> **Question-1**
>
> In Section 3.2, successful experience episodes are collected online during evaluation. Whenever a task is completed successfully, we store the corresponding plan (contact strategy and the cost function) together with the language description of the task and the estimated physical parameters. Later, when a similar task description and estimated parameters are encountered, the stored experience is retrieved and its associated plan is used as the initial plan for that episode.
>
> **Question-2**
>
> We have added the prompt used to generate the initial cost function in Appendix A.10 of the revised manuscript. The prompt used to generate the contact strategy was already included and remains available in Appendix A.12.
>
> **Question-3**
>
> In the Experiments section under Comparative Baselines, we now explicitly clarify how the baseline VLA models are initialized on the LIBERO dataset(lines 373-375) . Specifically, for each model we use the officially released LIBERO-OBJECT checkpoint for pick-and-place tasks and the LIBERO-GOAL checkpoint for all other tasks.
>
> **Question-4**
>
> We refer the reviewer to our response to the first weakness, where we provide a detailed definition of the action space. To provide immediate clarity: MPPI operates directly in a continuous action space (e.g., continuous end-effector deltas) and does not use motion primitives.
>
> MPPI optimizes trajectories to minimize a task-specific cost function. This cost function is dynamically generated (and refined) by the LLM based on interaction history. Importantly, the LLM is not selecting from a fixed library of primitives; rather, it composes mathematical cost terms based on available state, pose, and action variables. We have emphasized this flexibility in the revised manuscript (lines 234–237).
>
> Regarding the code implementation, could you kindly point us to the specific lines that gave the impression of motion primitives? We suspect this may be a misunderstanding of our cost-generation modules, and we would appreciate the opportunity to clarify the code or improve our documentation to prevent future confusion.

---

### Official Review · Reviewer_Tdgz · 2025-11-04

**Soundness:** 2
**Presentation:** 2
**Contribution:** 3
**Rating:** 4
**Confidence:** 4

**Summary:**

This work proposes a modular robotic system built with a combination of LLMs, VLMs, pose estimators and motion planners for manipulation task. Unlike what title may imply (VLAs), the modular system does not require demonstrations for imitation learning. Instead, it first uses perception modules such as FoundationPose to estimate poses of the objects and use VLMs for physical parameter specification, which get converted into a physics-based model (the simulator). LLM is used to provide/refine task-specific cost functions. Leveraging the model and the inferred cost functions, an MPC algorithm (MPPI) is used to generate robot actions. The proposed method is evaluated in Robosuite and compared against the OpenVLA and pi-0.5 baselines. Ablation experiments are also done to highlight the importance of the various components of the method.

**Strengths:**

- Leveraging the knowledge of foundation models in a reactive MPC formulation is an interesting and promising direction that bypasses the requirement for extensive demonstration data compared to imitation learning based approaches.
- The tasks demonstrated in this work, albeit in simulation, involve contact-rich interactions which are impressive to achieve without task-specific demonstrations.

**Weaknesses:**

- The term “real world” or “physical world” are mentioned several places in the paper. However, this is an incorrect claim as experiments are only done in simulation. This is particularly relevant in the context of this work, where high fidelity model is often required for such contact-rich reasoning, which often induces a notable gap between simulation and the real world (for both real2sim reconstruction and sim2real transfer). While it is fine to only demonstrate simulation results and state its limitations, leaving the incorrect impression that the proposed method works in the real world is certainly a over claim.
- Additionally, the method is named as “Contact-VLA”, which is also inaccurate. VLAs typically refer to end-to-end models finetuned from VLMs to directly produce actions. However, in this work, it is a modular approach that leverages a combination of VLMs and LLMs. As a result, it would also easily lead to incorrect expectation of the work.
- Since the approach involves using MPPI for planning and is demonstrated only in simulation, it is unclear whether MPPI uses the same simulation environment as that being used for evaluation. If so, it would not establish a fair comparisons to OpenVLA or pi-0.5 since they do not assume access to the environment models.
- The approach also appears to rely on a text-only LLM for specifying contact strategies and cost functions. This appears to be very challenging since it’s unclear how LLMs can specify these given no visual input (or even coordinate frame). The provided appendix also does not seem to provide sufficient details. For example, does it rely on task-specific prompt that is manually refined by the developer?

**Questions:**

See the weaknesses section.

---

> ### Author Response · Authors · 2025-11-21
> **Clarifying "Real World" Terminology, Renaming to CoRAL and Simulation Environments**
>
> **Weakness-1**
>
> **"Real World" Terminology:** By the phrase “real world” we intended to refer to the “evaluation world” (which, in our work, is itself a simulated physics environment). But, we acknowledge that this is confusing. We have therefore replaced all occurrences of “real world” with “evaluation world” throughout the paper to ensure terminological consistency and avoid further confusion. Our rationale is that MPPI operates by sampling trajectories in auxiliary “inner worlds” or “planning worlds,” which are initialized using information obtained from the evaluation world via the VLM and the pose tracker. These planning worlds are used solely for hypothetical rollouts and have no direct consequences; only the final selected action is executed in the evaluation world, which determines task success or failure.
>
> While we acknowledge the sim-to-real gap arising from unmodeled dynamics and sensing noise, our approach offers inherent resilience to these discrepancies. Specifically, the use of generalist Vision-Language Models (VLMs) helps bridge the perceptual gap by generalizing across both photorealistic and simulated domains for world building. Additionally, the reactive capabilities of the LLM allow the system to compensate for dynamic mismatches between real-world physics and the MuJoCo simulation. Thus, our architecture provides a strong foundation for deployment.
>
>
>
> **Weakness-2**
>
> Our rationale for using the phrase “VLA” was that our modular approach takes image and language inputs and utilizes a VLM and an LLM to process the respective modalities, even though it is not an end-to-end trained system. We fully understand your concern and the potential misunderstanding that the term “VLA” can cause, given its common association with end-to-end trained models. For this reason, we have changed the name of our model to “Contact-Rich Adaptive LLM-based control(CoRAL)” and no longer refer to it as a “Contact-VLA” in the revised manuscript.
>
> **Weakness-3**
>
> **Distinction of Evaluation and Planning Environments:** We consider two types of environments, both instantiated in the MuJoCo physics engine: an “evaluation environment” and a “planning environment.” MPPI uses the planning environment to roll out randomly sampled action trajectories, and only the action selected by MPPI is executed in the evaluation environment. Images sent to the VLM for physical parameter estimation and to FoundationPose for pose tracking are captured from the evaluation environment. At each iteration, the planning environment is initialized using the physical parameters estimated from the evaluation environment and the object poses returned by FoundationPose.
>
> In principle, the evaluation environment can be replaced by the actual physical world, since both the VLM queries and the pose tracking operate directly on images and do not rely on simulation-specific signals. In this case, the planning environments would remain simulated, and the primary remaining source of sim-to-real discrepancy would be the fidelity of the MuJoCo physics model to real-world dynamics. Our approach therefore assumes access to a standard physics engine (like MuJoCo). Under this assumption, we believe that comparisons to end-to-end VLA models are meaningful and fair.
>
> **Benefits of Using an Explicit Model:** End-to-end VLA models are expected to learn an implicit model of physical dynamics from their training data, whether exposed to physical reality (via real-world teleoperation) or a simulated equivalent (via sim teleoperation, as in our baselines). While these models theoretically possess some capacity to generalize to unseen tasks, our results in Table 1 demonstrate that this learned implicit model is insufficient for unseen contact-rich manipulation. Specifically, the VLA baselines achieve a success rate near 0% on these complex tasks. This failure indicates that current VLA architectures struggle to acquire a sufficiently robust internal model of physical dynamics. Our framework succeeds precisely because it leverages an explicit, adaptable physics model where data-driven implicit models fall short.
>
> **Upcoming Qualitative Evidence:** We will be adding rollout videos to qualitatively demonstrate that baseline failures stem from an inability to master complex contact dynamics. Please refer to Point 7 in our General Response for details.

---

> ### Author Response · Authors · 2025-11-21
> **LLM Strategy Generation**
>
> **Weakness-4**
>
> The approach also appears to rely on a text-only LLM for specifying contact strategies and cost functions. This appears to be very challenging since it’s unclear how LLMs can specify these given no visual input (or even coordinate frame). The provided appendix also does not seem to provide sufficient details. For example, does it rely on task-specific prompt that is manually refined by the developer?
>
> **LLM-input:** The LLM is not operating blindly. While the strategy generation is text-based, it is conditioned on structured information: specifically, the tracked 6D poses of relevant objects and the end-effector pose. In our experiments, this geometric context was sufficient for the LLM to synthesize successful contact strategies.
>
> **Prompt:** We have added the full prompt used for the initial cost function generation to Appendix A.10 for completeness. Previously, this prompt was not listed separately because it closely resembled the prompt used for cost refinement, and we mistakenly assumed the refinement prompt would be sufficient to convey the complete process. The VLM prompt for physical parameter estimation (Appendix A.9), the strategy refinement prompt (Appendix A.11.1), the world model correction prompt (Appendix A.11.2), and the contact strategy generation prompt (Appendix A.12) were already included in our original submission. All of these prompts follow a generic template in which the task description, tracked object poses, and estimated physical parameters can change dynamically.
>
> **Upcoming Expert Baseline:** We will be adding a "Handcrafted Multi-Phased Baseline" to demonstrate how the LLM-generated costs match expert human performance. Please see Point 7 in our General Response.

---

### Author Response · Authors · 2025-11-21
**General Response: Introducing CoRAL, Clarifying Terminology, and Re-stating Core Contributions,  Upcoming Manuscript Additions**

We thank the reviewers for their detailed feedback. We noticed several consistent misunderstandings regarding our terminology, experimental setup, and the scope of our contributions. To ensure clarity we address these high-level concerns here. The manuscript has been revised to reflect these changes.

**1. Renaming to CoRAL (Clarifying the Architecture):** Reviewers correctly pointed out that "Contact-VLA" implies an end-to-end trained model. To prevent this confusion and accurately reflect our modular architecture, we have renamed the method to **CoRAL (Contact-Rich Adaptive LLM-based control)**. We no longer use the term "VLA" to describe our method in the revised text.

**2. Clarification of "Real World" Terminology (Sim-to-Sim Verification):** We acknowledge the confusion and have replaced "real world" with "evaluation world".

* **Rationale:** We utilize two simulated environments: an internal planning world for MPPI rollouts and an external evaluation world where the agent executes actions, both within MuJoCo simulated environments.

* **Validity:** We treat the evaluation world as a proxy for reality. As our perception modules operate strictly on RGB images without privileged states, the pipeline is architecturally identical to a physical deployment.

**3. Continuous Action Space (No Motion Primitives):** CoRAL does **not** use motion primitives. It utilizes MPPI to optimize directly over a continuous 7-DoF action space. The LLM formulates the cost function, not skills.

**4. Explicit Novelty Statement: "LLM-as-a-Cost-Designer:"** We respectfully disagree with the assessment that our contribution is limited to architectural modularity. Our primary novelty is the formulation of **LLM-as-a-Cost-Designer**. We introduce a paradigm where an LLM translates semantic goals into dense, continuous cost landscapes for MPPI.

* **Why this matters:** This enables the zero-shot execution of complex, contact-rich behaviors (such as force regulation and pivoting) that end-to-end VLA models failed to learn in our benchmarks. The synergy of foundation models with physics-based planning, augmented by online parameter adaptation, offers a distinct and effective alternative to the data-heavy imitation learning paradigm.

**5. Sufficiency of Physical Parameters:** Concerns were raised regarding the sufficiency of estimating only mass and friction. For the rigid-body contact manipulation tasks we target (pushing, pivoting, flipping), mass and surface friction are the dominant dynamics parameters.

* **Robustness:** Crucially, our contribution is not the complexity of the initial estimate, but the adaptation mechanism. As shown in Figure 3, our system detects discrepancies and corrects these parameters online.

**6. Safety and Explainability vs. Black-Box VLAs:** While we acknowledge that LLM-based planners lack formal stability guarantees, this remains an open challenge for the entire field of generative AI. However, CoRAL offers superior safety characteristics compared to the baselines:

* **Transparency:** Unlike black-box VLAs, CoRAL exposes an explicit cost function that links behavior directly to optimization objectives.
* **Refinement:** Our reactive loop allows the system to critique and correct its own strategies based on physical feedback before catastrophic failure occurs. We argue that this explicit "reasoning visibility" makes CoRAL safer for deployment than opaque end-to-end policies.

**7. Upcoming Manuscript Additions: New Baselines & Visual Evidence** To further validate our claims and allow for a direct comparison with human expertise, we are currently finalizing the following additional analyses which will be incorporated into the manuscript shortly:

* **Handcrafted Cost Baselines (Expert vs. LLM):** A key feature of CoRAL is the LLM's ability to propose cost functions with a finite state machine (FSM) structure (e.g., decomposing the "Push and Pick Cutting Board" task into a sliding phase and a grasping phase for). To benchmark this, we are working on two new baselines:

    * **Handcrafted Multi-Phased Baseline:** An expert-designed cost function utilizing the same multi-phase logic. This will demonstrate the effectiveness of the LLM as a cost generator comparable to human experts.
    * **Handcrafted Single-Phase Baseline:** An expert-designed cost function without phase decomposition. This will highlight the necessity of the high-level FSM strategy and the difficulty of solving these tasks with a single, monolithic cost function.
* **Rollout Videos:** We are also preparing rollout videos of the baseline VLAs. These will serve two purposes: first, to visually demonstrate that the tasks involve complex dynamics that are non-trivial, refuting potential concerns about task simplicity; and second, to showcase the qualitative limitations of end-to-end VLAs, specifically highlighting their inability to master these contact-rich behaviors compared to CoRAL's successful executions.

---

> ### Author Response · Authors · 2025-11-27
> **Empirical Validation of Task Complexity: Qualitative Rollouts & New Expert Baselines**
>
> **Comparison of the Rollouts of OpenVLA-OFT and CoRAL**
>
> To address concerns regarding task complexity and the necessity of explicit physical reasoning, we provide a qualitative comparison between CoRAL and the OpenVLA-OFT baseline (initialized with the LIBERO-Goal checkpoint). We analyze two tasks that strictly require contact-rich dynamics reasoning: Push with Constant Force and Flip with Wall. We provide the rollout at:
> https://youtube.com/playlist?list=PLnSB3c1by2MNjZKRzR9X4eDDBYaCOaPxB&si=0Xw8HdmOwwXj5jd9
>
> **1. Push with Constant Force**
>
> We initialize the scene with a low-mass box, making it highly susceptible to tipping if the contact point creates a large moment arm.
>
> The baseline VLA applies force above the center of mass. This creates a large moment arm and excessive torque, causing the object to rotate and tip rather than a consistent slide. This illustrates that the VLA lacks an understanding of the object's inertial properties and the relationship between contact height and rotational stability.
>
> CoRAL estimates and refines the mass via the VLM/LLM pipeline and generates a contact strategy that aligns the push with the object's center of mass. MPPI then optimizes for a central contact point, pushing the object while maintaining stability. This illustrates CoRAL’s ability to ground semantic goals in physical reality, explicitly coupling motion planning with force and stability constraints.
>
> **2. Flip with Wall**
>
> The VLA executes the push phase, establishing contact with the wall. However, it pushes the box into a configuration where the contact region required for the flip becomes inaccessible. Additionally, upon reaching the wall, the policy stagnates, attempting no flipping maneuver, though the flip was likely already rendered infeasible by this misalignment.
>
> CoRAL executes a multi-stage strategy. First, it pushes the box until stable contact with the wall is achieved. Only then does it transition to a pivoting motion. Crucially, the video shows CoRAL repositioning the end-effector to a new contact region to generate the necessary torque, successfully using the wall as a tool to flip the box.
>
> These qualitative results reinforce that the proposed tasks are not trivial. They require an understanding of environmental constraints that SOTA end-to-end models (like OpenVLA) do not possess. CoRAL succeeds specifically because it decouples semantic reasoning (Strategy/Cost) from low-level dynamics control (MPPI).
>
> **Human Expert Baselines**
>
> To address concerns regarding task complexity and compare LLM cost generation against a human expert, we introduced two new human-expert baselines:
>
>   * Single-Stage Expert (monolithic cost function)
>  * Multi-Phase Expert (FSM-structured cost function)
>
> Table 1 details the new baseline results.
>
>  **Empirical Validation of Task Complexity:** Reviewers questioned if the tasks were "very simple". Our results refute this.
>
> * The Failure of Monolithic Objectives: As shown in the Expert (Single-Stage) row, even a human-tuned cost function fails completely on T1 (Push+Pick Board) (0/10 success) and performs poorly on T6 (Flip w/ Wall) (3/10 success).
>
> * The Necessity of Phasing: Success is only recovered when the expert explicitly decomposes the problem into distinct phases (e.g., sliding vs. grasping), as seen in the Expert (FSM) results (jumping to 8/10 for T1 and 9/10 for T6).
> This gap demonstrates that these tasks possess inherent temporal and semantic complexity unsolvable by simple trajectory optimization alone. They strictly require the high-level logic and phase transitions CoRAL provides, confirming the tasks are non-trivial benchmarks for contact-rich manipulation.
>
> **Comparison of LLM-Generated vs. Expert-Designed Costs:** Reviewers asked whether an LLM, relying on text without visual feedback, can generate effective optimization objectives.
>
> We treat the Expert (FSM) baseline as an empirical "upper bound" for performance given the MPPI controller. By comparing CoRAL against this oracle, we observe that the LLM-generated strategies are highly competitive, though they are naturally behind the manually tuned expert designed strategies. Despite a gap in the hardest tasks (e.g., T1, where CoRAL achieves 4/10 vs. the expert's 8/10), the results demonstrate that the LLM successfully synthesizes the high-level logic (phase transitions) that the Expert (Single-Stage) baseline failed to capture. While CoRAL does not fully match the reliability of the hand-tuned expert on the hardest tasks, it effectively bridges the gap between semantic intent and control, eliminating the manual design needed for the Expert FSM results.
>
> We attribute the remaining gap to the limitations of the generalist GPT-4o model. We believe fine-tuning a specialized LLM for physics reasoning could bridge this final gap, potentially boosting performance to expert levels, an exciting direction for future research.

---

> ### Author Response · Authors · 2025-11-27
> **Summary of Manuscript Revisions**
>
> * Replaced all instances of "real world" with "evaluation world" to ensure terminological consistency (Throughout Manuscript and the Figures).
> * Renamed the model and the title from "Contact-VLA" to "Contact-Rich Adaptive LLM-based control (CoRAL)" (Throughout Manuscript).
> * Added the full prompt used for the initial cost function generation (Appendix A.10).
> * Clarified initialization of baseline VLA models using LIBERO-OBJECT and LIBERO-GOAL checkpoints (lines 373-375).
> * Fixed formatting issue regarding Figure-3.
> * Revised Equation (1) and added definitions for $J_{\textrm{mem}}$ and $C_{\textrm{mem}}$ (lines 214-215).
> * Extended the discussion on strategies to mitigate computational latency (lines 778-791).
> * Added specific breakdown of end-to-end latency sources (lines 792-802).
> * Clarified that the cost function expression is illustrative and the LLM is not restricted to a fixed set of cost terms (lines 234-237).
> * Emphasized that object masses, surface friction coefficients, and object dimensions vary randomly between the initialization of the environments (lines 336-337).
> * Added "Human Expert-Designed Cost" baselines (single-stage and FSM) to the Comparative Baselines (Table-1 and lines 376-394).
> * Added a new results subsection comparing CoRAL to Human-Designed Cost Functions (lines 418-428).

---

### Author Response · Authors · 2025-11-27
**Concluding Remarks and Invitation to Re-evaluate**

We sincerely thank the reviewers for their time and constructive feedback.

We acknowledge that our initial choice of terminology, specifically the use of "real world" and the name "Contact-VLA", led to unintended confusion regarding our experimental setup and the scope of our contributions. We have carefully revised the manuscript to resolve these ambiguities. Notably, we have renamed our method to CoRAL (Contact-Rich Adaptive LLM-based control) and clarified our evaluation protocols to better distinguish our work from end-to-end VLA approaches.

We believe that our detailed point-by-point responses, combined with the clarifications in the revised manuscript, effectively address the concerns regarding novelty, baselines, and task complexity. In light of these updates and the additional evidence provided, we respectfully invite you to re-evaluate our submission.

If you have any remaining questions or major concerns, we are eager to address them promptly before the discussion period ends.

---

### Author Response · Authors · 2025-12-04
**Summary of Rebuttal and Context for Re-evaluation**

Dear Area Chair,

Given the recent incident and review reversion, we are writing to summarize our rebuttal to assist your assessment. Notably, we received no reviewer engagement prior to the reset; thus, our current responses contain all the relevant updates and clarifications without any missing discussion.We respectfully ask that you weigh these substantial revisions heavily when making the final decision.

We strongly believe the initial low ratings were driven primarily by specific, remediable presentation issues rather than fundamental methodological flaws. The reviewers expressed confusion over our terminology (specifically the use of "Real World" for simulation and "VLA" for our modular system) and flagged a formatting oversight. We have addressed these entirely in the revision: we renamed our method to CoRAL, adopted strict "Evaluation World" terminology to avoid confusion and corrected the formatting issue. Furthermore, to address concerns regarding task complexity, we introduced new Human Expert Baselines (Single-Phase and Multi-Phase). These new empirical results demonstrate that the tasks are non-trivial and that our LLM-generated costs perform competitively with expert-designed logic, directly refuting the criticism that the tasks were too simple.

Despite the low quantitative scores, the content of the initial reviews confirms the soundness of our approach. Reviewer 1 explicitly noted that leveraging foundation models in this way is a "promising direction that bypasses the requirement for extensive demonstration data" and found the contact-rich behaviors "impressive." Reviewer 2 agreed that using LLMs to write cost functions is "interesting" and confirmed our experiments showed the "effectiveness of each module." Similarly, Reviewer 4 highlighted that our framework addresses "a significant challenge in robotics" with "a clear performance advantage," while Reviewer 3 acknowledged that our method demonstrates "superior performance compared to baseline approaches."

Given that we have resolved the specific presentation and terminology issues that led to the initial rejection ratings, and considering the reviewers' own consensus on the promise and performance of the method, we believe the paper merits a score significantly higher than the initial assessment. We recognize the additional workload this situation places on you, and we sincerely appreciate your time in taking a fresh look at our revised manuscript and updated baselines to render a final decision based on the paper's current technical standing.

---

### Meta-Review · Area_Chair_FV3G · 2025-12-27

**Summary:**

This paper proposes a framework that integrates LLMs with MPC, specifically using LLMs to generate and refine cost functions for MPPI control in robotic manipulation tasks. The approach is evaluated primarily in simulation on tasks from the LIBERO benchmark, along with several custom manipulation scenarios. The authors report that their method outperforms end-to-end VLA baselines such as OpenVLA and π0.5 across a subset of tasks.

While reviewers generally find the high-level idea of incorporating LLMs for symbolic reasoning within an MPC framework to be interesting and timely, the overall assessment of the paper remains negative. The primary concerns include inaccurate positioning of the contribution relative to prior work, insufficient comparison with existing LLM-based planning and control systems, and the simplicity of the evaluation tasks. Although the authors have promptly addressed the positioning of the work, the revised framing necessitates comparisons with more appropriate and representative baselines. Consequently, I recommend rejection, but believe that incorporating additional baselines and evaluating on more challenging benchmarks would substantially strengthen the paper and more convincingly substantiate the claimed advantages of the proposed approach.

**Reviewer Concerns:**

A major concern raised across reviews is that the initial version of the paper mischaracterized the proposed system as a VLA approach. Reviewers emphasized that the method does not involve imitation learning or end-to-end policy learning, but instead constitutes an LLM-guided planning-and-control pipeline. Although the authors partially addressed this issue by renaming and repositioning the approach in the rebuttal, reviewers noted that the original framing undermined clarity and created confusion about the paper’s technical contribution. Similarly, the use of the term “real-world” was considered misleading, as all experiments were conducted in simulation. The authors have devoted much effort to address these issues during the rebuttal phase.

Reviewers expressed concerns that the evaluated tasks are relatively simple compared to the paper’s claims of “contact-rich” manipulation. Many tasks involve limited or structured contacts and do not convincingly demonstrate robustness in complex contact scenarios. Additionally, the approach assumes access to accurate object geometry and known object models, which reviewers felt significantly weakens the generality of the claims, especially when contrasted with learning-based baselines designed to operate under more realistic perception uncertainty.

Another recurring concern is the choice of baselines. While comparisons to OpenVLA and pi-0.5 are informative, reviewers noted that these systems are trained end-to-end policies optimized for different objectives and computational regimes. Since the proposed method relies on online planning, MPC rollouts, and repeated LLM calls, reviewers argued that comparisons would be more meaningful if they included other LLM-based planning or MPC-style methods, rather than exclusively imitation-learning-based VLAs.

One reviewer specifically highlighted that computational cost, inference latency, and planning overhead are not sufficiently discussed or incorporated into the evaluation. Given that the method relies on MPPI rollouts and intermittent LLM inference, reviewers felt that ignoring runtime considerations makes the comparison to reactive VLA policies incomplete and potentially unfair.

The weaknesses of the the empirical results (baselines + tasks) were understandably not addressed during the rebuttal phase.

**Reviewer Scores:**

During rebuttal the authors updated the manuscript to address issues around misuse and overloaded use of terminology and positioning of the method, which could justify modest score increases for some reviewers. However, the core technical concerns were not fully addressed in the authors’ response. As a result, it is unlikely that reviewer scores would substantially increase.

---

### Decision · Program_Chairs · 2026-01-26

Reject